# ErrorCompensatedX: error compensation for variance reduced algorithms

**Hanlin Tang**
Department of Computer Science
University of Rochester
tanghl1994@gmail.com

**Yao Li**
Department of Mathematics
Michigan State University
liyao6@msu.edu

**Ji Liu**
Kuaishou Technology
ji.liu.uwisc@gmail.com

**Ming Yan**
Department of Computational Mathematics, Science
and Technology; Department of Mathematics
Michigan State University
myan@msu.edu

## Abstract

Communication cost is one major bottleneck for the scalability for distributed learning. One approach to reduce the communication cost is to compress the gradient during communication. However, directly compressing the gradient decelerates the convergence speed, and the resulting algorithm may diverge for biased compression. Recent work addressed this problem for stochastic gradient descent by adding back the compression error from the previous step. This idea was further extended to one class of variance reduced algorithms, where the variance of the stochastic gradient is reduced by taking a moving average over all history gradients. However, our analysis shows that just adding the previous step's compression error, as done in existing work, does not fully compensate the compression error. So, we propose ErrorCompensatedX, which uses the compression error from the previous two steps. We show that ErrorCompensatedX can achieve the same asymptotic convergence rate with the training without compression. Moreover, we provide a unified theoretical analysis framework for this class of variance reduced algorithms, with or without error compensation.

## 1   Introduction

Data compression reduces the communication volume and alleviates the communication overhead in distributed learning. E.g., Alistarh et al. (2017) compress the gradient being communicated using quantization and find that reducing half of the communication size does not degrade the convergence speed. However, the convergence speed would be slower if we further reduce the communication size, and it requires the compression to be unbiased (Alistarh et al., 2017; Tang et al., 2019). Alternatively, recent work (Stich et al., 2018a) shows that compression with error compensation, which adds back the compression error to the next round of compression, using only $3\%$ of the original communication volume does not degrade the convergence speed, and it works for both biased and unbiased compression operators.

Despite the promising performance of error compensation on stochastic gradient descent (SGD), SGD admits a slow convergence speed if the stochastic gradient has a large variance. Variance reduction techniques, such as **Momentum SGD** (Zhang et al., 2015), **ROOT-SGD** (Li et al., 2020), **STORM** (Cutkosky & Mehta, 2020), and **IGT** (Arnold et al., 2019), are developed, and they admit increased convergence speeds either theoretically or empirically. We found that directly applying

35th Conference on Neural Information Processing Systems (NeurIPS 2021).

error compensation to those variance reduced algorithms is not *optimal*, and their convergence speeds degrade. So a natural question arises: **what is the best compression method for variance reduced algorithms?** In this paper, we answer this question and propose ErrorCompensatedX, a general method for error compensation, and show that it admits faster convergence speeds than previous error compensation methods. The contributions of this paper can be summarized as follows:

- We propose a novel error compensation algorithm for variance reduced algorithms and SGD. Our algorithm admits a faster convergence rate compared to previous methods (Zheng et al., 2019a; Stich et al., 2018b) by fully compensating all error history.

- We provide a general theoretical analysis framework to analyze error compensated algorithms. More specifically, we decompose the convergence rate into the sum of two terms

$$\frac{1}{T} \sum_{t=1}^{T} \|\nabla f(\boldsymbol{x}_t)\|^2 \leq \mathcal{R}_{\text{uncompressed}} + \mathcal{R}_{\epsilon},$$

where $\mathcal{R}_{\text{uncompressed}}$ depends only on the convergence rate of the original algorithm without compression and $\mathcal{R}_{\epsilon}$ is depends only on the magnitude of the compression error $\epsilon$. It means that we can easily attain the convergence rate for any compressed algorithm in the form of (2). To the best of our knowledge, this is the first general result for error compensation.

## 2   Related Works

**Variance Reduced Optimization.**   When the variance of the stochastic gradient is large, training using SGD becomes unstable, and it usually renders a slow convergence speed. Many studies try to reduce the variance of the stochastic gradient. For example, Momentum SGD (Zhang et al., 2015) takes a moving average over all previous stochastic gradients. However, its expected momentum does not equal to the full gradient. Recent studies take one step further by applying bias correction to the momentum. One approach is to compute the gradient at $\boldsymbol{x}_t$ and $\boldsymbol{x}_{t-1}$ using the same data sample $\xi_t$ (Huang et al., 2020; Yuan et al., 2020; Li et al., 2020), while another direction is to compute the gradient at the extrapolated point of $\boldsymbol{x}_t$ and $\boldsymbol{x}_{t-1}$ (Arnold et al., 2019; Cutkosky & Mehta, 2020). When the number of data samples is finite, some approaches compute the full gradient once in a while or memorize the latest stochastic gradient of each data sample and use its average to approximate the full gradient; examples include SVRG (Johnson & Zhang, 2013), SAGA (Defazio et al., 2014), and SARAH (Nguyen et al., 2017).

**Communication Efficient Optimization.**   One major bottleneck of distributed learning is the communication cost. Methods, such as data compression, decentralized training, local SGD, and federated learning, were proposed to reduce the communication cost. Data compression is an important approach that can be combined with other approaches, and it is critical for communication networks with limited bandwidth. It was firstly proposed in Seide et al. (2014), where authors proposed **1-bit SGD**, which uses one bit to represent each element in the gradient but still achieves almost the same convergence speed with the uncompressed one. In Stich et al. (2018b), authors proposed a general algorithm for error compensated compression—**MEM-SGD** with theoretical analysis. They found that **MEM-SGD** has the same asymptotic convergence rate with uncompressed SGD, and more importantly, is robust to both biased and unbiased compression operators. This method can be combined with decentralized training (Vogels et al., 2020), local SGD (Xie et al., 2020), and accelerated algorithms (Gorbunov et al., 2020). Due to its promising efficiency, error compensation has been applied into many related area (Zheng et al., 2019b; Phuong & Phong, 2020; Yu et al., 2019; Shi et al., 2019; Ivkin et al., 2019; Sun et al., 2019; Basu et al., 2019; Vogels et al., 2019) to reduce the communication cost.

Another direction to minimize the side-effect of compression is to compress the difference between the current gradient and momentum. Previous work (Mishchenko et al., 2019; Liu et al., 2020) proved that this strategy achieves linear convergence when the loss function is strongly convex. However, the linear convergence requires either the full deterministic gradient, or using SVRG (Horváth et al., 2019) to control the variance when for finite sum the loss function. Therefore how they perform with general variance reduced stochastic algorithms is still an open problem.

# 3 Algorithm Design

## 3.1 Background

The underlying problem we consider can be posed as the following distributed optimization problem:

$$\min_{\boldsymbol{x}} \quad f(\boldsymbol{x}) = \frac{1}{n} \sum_{i=1}^{n} \underbrace{\mathbb{E}_{\xi^{(i)} \sim \mathcal{D}_i} F\left(\boldsymbol{x}; \xi^{(i)}\right)}_{:=f_i(\boldsymbol{x})}, \tag{1}$$

where $n$ is the number of workers, $\mathcal{D}_i$ is the local data distribution for worker $i$ (in other words, we do not assume that all nodes can access the same data set), and $F(\boldsymbol{x}; \xi^{(i)})$ is the local loss function of model $\boldsymbol{x}$ given data $\xi^{(i)}$ for worker $i$.

One widely used method for solving (1) is SGD, which updates the model using $\boldsymbol{x}_{t+1} = \boldsymbol{x}_t - \gamma \boldsymbol{g}_t$, where $\boldsymbol{x}_t$ is the model at the $t$-th iteration, $\gamma$ is the learning rate and $\boldsymbol{g}_t$ is the averaged stochastic gradients. However, SGD suffers from the problem of large stochastic gradient variance, and many variance reduced algorithms are developed. In our paper, we focus on the following variance reduced algorithms: Momentum SGD, STROM, ROOT-SGD[1], and IGT, because they all construct the estimator using a moving average of previous stochastic gradients, which can be summarized as

$$\boldsymbol{v}_t = (1 - \alpha_t) \boldsymbol{v}_{t-1} + \alpha_t \mathscr{A}(\boldsymbol{x}_t; \xi_t), \quad \boldsymbol{x}_{t+1} = \boldsymbol{x}_t - \gamma \boldsymbol{v}_t, \tag{2}$$

where $\boldsymbol{v}_t$ is the gradient estimator, $\mathscr{A}(\boldsymbol{x}_t; \xi_t)$ is a variable that depends on the history models $\boldsymbol{x}_s$ (for all $s \leq t$) and data sample $\xi_t$, and $\alpha_t$ is a scalar. Notice that $\mathscr{A}(\boldsymbol{x}_t; \xi_t)$ and $\alpha_t$ are designed differently for different algorithms. We list the different choices of those parameters for each algorithm in Table 1.

| | $\alpha_t$ | $\mathscr{A}(\boldsymbol{x}_t; \xi_t)$ |
|---|---|---|
| SGD | 1 | $\nabla F(\boldsymbol{x}_t; \xi_t)$ |
| Momentum SGD | $\alpha$ | $\nabla F(\boldsymbol{x}_t; \xi_t)$ |
| STORM | $\alpha$ | $\frac{1}{\alpha_t}\left(\nabla F(\boldsymbol{x}_t; \xi_t) - (1 - \alpha_t)\nabla F(\boldsymbol{x}_{t-1}; \xi_t)\right)$ |
| ROOT-SGD | $1/t$ | $\frac{1}{\alpha_t}\left(\nabla F(\boldsymbol{x}_t; \xi_t) - (1 - \alpha_t)\nabla F(\boldsymbol{x}_{t-1}; \xi_t)\right)$ |
| IGT | $\alpha$ | $\nabla F\left(\boldsymbol{x}_t + \frac{1-\alpha_t}{\alpha_t}(\boldsymbol{x}_t - \boldsymbol{x}_{t-1}); \xi_t\right)$ |

Table 1: Different choices of $(\alpha_t, \mathscr{A}(\boldsymbol{x}_t; \xi_t))$ for each variance reduced algorithm.

Recent work suggests that instead of compressing the gradient directly (Alistarh et al., 2017), using error compensation (Stich et al., 2018b) could potentially improve the convergence speed. The idea of error compensation is quite straightforward: adding back the compression error from the previous step to compensate the side-effect of compression. Denoting $C_\omega[\cdot]$[2] as the compressing operator, the updating rule of this method follows

$$\boldsymbol{x}_{t+1} = \boldsymbol{x}_t - \gamma C_\omega[\boldsymbol{g}_t + \boldsymbol{\delta}_{t-1}], \quad \boldsymbol{\delta}_t = \boldsymbol{g}_t + \boldsymbol{\delta}_{t-1} - C_\omega[\boldsymbol{g}_t + \boldsymbol{\delta}_{t-1}],$$

where $\boldsymbol{\delta}_t$ is the compression error at the $t$-th step. Moreover, recent works (Chen et al., 2020; Wu et al., 2018) find that adding a low-pass filter to the history compression error could be helpful for stabilizing the training and further improving the training speed. Their updating rule can be summarized as

$$\boldsymbol{e}_t = (1 - \beta)\boldsymbol{e}_{t-1} + \beta \boldsymbol{\delta}_{t-1}, \quad \boldsymbol{x}_{t+1} = \boldsymbol{x}_t - \gamma C_\omega[\boldsymbol{g}_t + \boldsymbol{e}_t], \quad \boldsymbol{\delta}_t = \boldsymbol{g}_t + \boldsymbol{e}_t - C_\omega[\boldsymbol{g}_t + \boldsymbol{e}_t],$$

where $\beta$ is a hyper-parameter of the low-pass filter.

Notice that even error compensated compression has been proved to be very powerful for accelerating the compressed training, it was only studied for SGD, which is a special case of (2) when $\alpha_t = 1$. For the case where $\alpha_t < 1$, previous work (Zheng et al., 2019a; Zhao et al., 2019; Wang et al., 2020) adapted the same idea from Stich et al. (2018b), which directly compresses the gradient estimator

---

[1]STORM (for non-convex loss function) and ROOT-SGD (strongly-convex loss function) use the same updating rule except $\alpha_t = 1/T^{\frac{2}{3}}$ for STORM and $\alpha_t = 1/T$ for ROOT-SGD.

[2]Here $\omega$ denotes the randomness of the compression operator.

$\boldsymbol{v}_t$ with the compression error from the previous step being compensated (we refer this method as **Single Compensation**). However, we find that when $\alpha_t$ is very small ($\mathcal{O}(1/T)$ for ROOT-SGD and $\mathcal{O}\left(1/T^{\frac{2}{3}}\right)$ for STORM, where $T$ is the total training iterations), Single Compensation would be much slower than the uncompressed one, as shown in Figure 1.

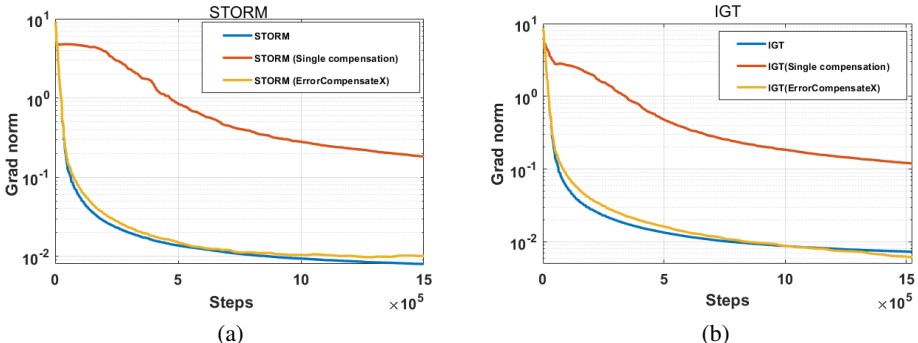

(a)  (b)

Figure 1: Convergence speed comparison on linear regression for STORM and IGT with different compression techniques. The $x$-axis is the training step number, and the $y$-axis is the norm of the full gradient. The batch size equals 1, $\alpha_t = 1/t$, and we use 1-bit compression as described in Tang et al. (2019). Single Compensation means only compensate the compression error from the last step, and we can see that it admits a much slower convergence speed than the uncompressed one when $\alpha_t$ is very small. However, our proposed ErrorCompensatedX has a similar convergence rate as the uncompressed one.

## 3.2 Proposed ErrorCompensatedX

In this section, we provide the intuition behind our proposed method ErrorCompensatedX. For simplicity, we consider the general update in (2) with $\alpha_t = \alpha$. Let $\boldsymbol{v}_{-1} = 0$, then the uncompressed algorithm has the update

$$\boldsymbol{x}_T = \boldsymbol{x}_0 - \gamma \sum_{t=0}^{T-1} \sum_{s=0}^{t} \alpha(1-\alpha)^{t-s} \mathscr{A}(\boldsymbol{x}_s; \xi_s).$$

Note $\mathscr{A}(\boldsymbol{x}_s; \xi_s)$ is transferred from worker nodes to the server (here we aggregate the data from all workers). If the compression is applied on $\mathscr{A}(\boldsymbol{x}_s; \xi_s)$, then we have the update

$$\begin{aligned}
\boldsymbol{x}_T =& \boldsymbol{x}_0 - \gamma \sum_{t=0}^{T-1} \sum_{s=0}^{t} \alpha(1-\alpha)^{t-s}(\mathscr{A}(\boldsymbol{x}_s; \xi_s) - \boldsymbol{\delta}_s) \\
=& \boldsymbol{x}_0 - \gamma \sum_{t=0}^{T-1} \sum_{s=0}^{t} \alpha(1-\alpha)^{t-s} \mathscr{A}(\boldsymbol{x}_s; \xi_s) + \gamma \sum_{s=0}^{T-1} (1-(1-\alpha)^{T-s}) \boldsymbol{\delta}_s.
\end{aligned}$$

Note that the sequence $\{\boldsymbol{x}_s\}_{s=0}^{T}$ is different from that of the uncompressed one. Here $\boldsymbol{\delta}_s$ is the compression error, and the compressed data $\mathscr{A}(\boldsymbol{x}_s; \xi_s) - \boldsymbol{\delta}_s$ is transferred to the server. With error compensation and additional $\boldsymbol{\delta}_{-1} = 0$, we have

$$\begin{aligned}
\boldsymbol{x}_T =& \boldsymbol{x}_0 - \gamma \sum_{t=0}^{T-1} \sum_{s=0}^{t} \alpha(1-\alpha)^{t-s}(\mathscr{A}(\boldsymbol{x}_s; \xi_s) + \boldsymbol{\delta}_{s-1} - \boldsymbol{\delta}_s) \\
=& \boldsymbol{x}_0 - \gamma \sum_{t=0}^{T-1} \sum_{s=0}^{t} \alpha(1-\alpha)^{t-s} \mathscr{A}(\boldsymbol{x}_s; \xi_s) + \gamma \sum_{s=0}^{T-1} \alpha(1-\alpha)^{T-1-s} \boldsymbol{\delta}_s.
\end{aligned}$$

Here $\boldsymbol{\delta}_s$ is the compression error that occurs while compressing $\mathscr{A}(\boldsymbol{x}_s; \xi_s) + \boldsymbol{\delta}_{s-1}$. When $\alpha = 1$, the last term disappears, and the standard SGD case has been analyzed in Tang et al. (2019); Stich et al.

(2018a). In this paper, we focus on the case where the last term does not disappear, i.e., $\alpha \in (0, 1)$. Given $\boldsymbol{\delta}_{-1} = \boldsymbol{\delta}_{-2} = 0$, when using ErrorCompensatedX, we have

$$
\begin{aligned}
\boldsymbol{x}_T =& \boldsymbol{x}_0 - \gamma \sum_{t=0}^{T-1} \sum_{s=0}^{t} \alpha(1-\alpha)^{t-s}(\mathscr{A}(\boldsymbol{x}_s; \xi_s) + (1-\alpha)(\boldsymbol{\delta}_{s-1} - \boldsymbol{\delta}_{s-2}) + \boldsymbol{\delta}_{s-1} - \boldsymbol{\delta}_s) \\
=& \boldsymbol{x}_0 - \gamma \sum_{t=0}^{T-1} \sum_{s=0}^{t} \alpha(1-\alpha)^{t-s}\mathscr{A}(\boldsymbol{x}_s; \xi_s) + \alpha\gamma\boldsymbol{\delta}_{T-1}.
\end{aligned}
$$

Notice that in the discussion above, we make an implicit assumption that there is only one worker in the training. For the case of multiple workers, we adapt the same strategy proposed by Tang et al. (2019), which uses a Parameter-Server communication prototype and applies error compensation for both rounds of worker-server communication.

For changing $\alpha_t$, the updating rule of ErrorCompensatedX with a low-pass filter follows

$$
\boldsymbol{e}_t = (1-\beta)\boldsymbol{e}_{t-1} + \beta\left(\frac{\alpha_{t-1}}{\alpha_t}(2-\alpha_t)\boldsymbol{\delta}_{t-1} - \frac{\alpha_{t-2}}{\alpha_t}(1-\alpha_t)\boldsymbol{\delta}_{t-2}\right), \quad \Delta_t = \mathscr{A}(\boldsymbol{x}_t; \xi_t) + \boldsymbol{e}_t,
$$

(3)

$$
\boldsymbol{v}_t = (1-\alpha_t)\boldsymbol{v}_{t-1} + \alpha_t C_\omega[\Delta_t], \quad \boldsymbol{\delta}_t = \Delta_t - C_\omega[\Delta_t], \quad \boldsymbol{x}_{t+1} = \boldsymbol{x}_t - \gamma\boldsymbol{v}_t.
$$

Here $\boldsymbol{e}_t$ is the compression error being passed after the low-pass filter and $\beta$ is the hyper-parameter of the low-pass filter. Our analysis indicates that for a general case with $0 < \alpha_t \le 1$, using just the compression error from the last step is not good enough, especially when $\alpha_t$ is very small. In this case, ErrorCompensatedX admits a much faster convergence speed than Single Compensation and could achieve the same asymptotic speed with the uncompressed one. We assume a Parameter-Server communication prototype for the parallel implementation of ErrorCompensatedX, and the detailed algorithm description can be find in Algorithm 1.

---

**Algorithm 1** ErrorCompensatedX for general $\mathscr{A}(\boldsymbol{x}; \xi)$

---

1: **Input:** Initialize $\boldsymbol{x}_0$, learning rate $\gamma$, loss-pass filter parameter $\beta$, initial error on workers $\boldsymbol{\delta}_{-1}^{(i)} = \boldsymbol{\delta}_0^{(i)} = 0$, initial error on the server $\boldsymbol{\delta}_{-1} = \boldsymbol{\delta}_0 = 0$, scheduler of $\{\alpha_t\}_{t=-1}^{T}$, and number of total iterations $T$. The initial gradient estimator $\boldsymbol{v}_0 = \nabla F\left(\boldsymbol{x}_0; \xi_0^{(i)}\right)$ with total batch-size $B_0$, initial compression error buffer $\boldsymbol{e}_0^{(i)} = 0$ and $\boldsymbol{e}_0 = 0$.
2: **for** $t = 1, 2, \cdots, T-1$ **do**
3:     **On worker** $i$:
4:         Pass the worker compression error into the low-pass filter: $\boldsymbol{e}_t^{(i)} = (1-\beta)\boldsymbol{e}_{t-1}^{(i)} + \beta\left(\frac{\alpha_{t-1}}{\alpha_t}(2-\alpha_t)\boldsymbol{\delta}_{t-1}^{(i)} - \frac{\alpha_{t-2}}{\alpha_t}(1-\alpha_t)\boldsymbol{\delta}_{t-2}^{(i)}\right)$.
5:         Compute the error-compensated local gradient estimator: $\Delta_t^{(i)} = \mathscr{A}\left(\boldsymbol{x}_t; \xi_t^{(i)}\right) + \boldsymbol{e}_t^{(i)}$.
6:         Compress $\Delta_t^{(i)}$ into $C_\omega\left[\Delta_t^{(i)}\right]$ and update the local worker error $\boldsymbol{\delta}_t^{(i)} \leftarrow \Delta_t^{(i)} - C_\omega\left[\Delta_t^{(i)}\right]$.
7:         Send $C_\omega\left[\Delta_t^{(i)}\right]$ to the parameter server.
8:     **On parameter server:**
9:         Pass the server compression error into the low-pass filter: $\boldsymbol{e}_t = (1-\beta)\boldsymbol{e}_{t-1} + \beta\left(\frac{\alpha_{t-1}}{\alpha_t}(2-\alpha_t)\boldsymbol{\delta}_{t-1} - \frac{\alpha_{t-2}}{\alpha_t}(1-\alpha_t)\boldsymbol{\delta}_{t-2}\right)$.
10:         Average all gradient estimator received from workers: $\Delta_t = \frac{1}{n}\sum_{i=1}^{n} C_\omega\left[\Delta_t^{(i)}\right] + \boldsymbol{e}_t$.
11:         Compress $\Delta_t$ into $C_\omega[\Delta_t]$ and update the server error $\boldsymbol{\delta}_t = \Delta_t - C_\omega[\Delta_t]$
12:         Send $C_\omega[\Delta_t]$ to workers
13:     **On worker** $i$:
14:         Update the gradient estimator $\boldsymbol{v}_t = (1-\alpha_t)\boldsymbol{v}_{t-1} + \alpha_t C_\omega[\Delta_t]$.
15:         Update the model $\boldsymbol{x}_{t+1} = \boldsymbol{x}_t - \gamma\boldsymbol{v}_t$.
16: **end for**
17: **Output:** $\boldsymbol{x}_T$

---

Throughout this paper and its supplementary document, we let $f^* = \min_{\boldsymbol{x}} f(\boldsymbol{x})$ be the optimal objective value. In addition, we let $\| \cdot \|$ denote the $l_2$ norm of vectors, and $\lesssim$ means "less than or equal to up to a constant factor".

# 4 Theoretical Analysis

In this section, we first summarize the general updating rule for each algorithm w/o error compensation. Then we will present a general theorem that is essential for getting the convergence rate of each algorithm. At last, we will see that by using ErrorCompensatedX, we could achieve the same asymptotic convergence rate with training without communication compression using both biased and unbiased compression.

All error compensation algorithms mentioned so far can be summarized into the formulation of

$$\boldsymbol{e}_t = (1 - \beta)\boldsymbol{e}_{t-1} + \beta \left( c_{1,t}\boldsymbol{\delta}_{t-1} - c_{2,t}\boldsymbol{\delta}_{t-2} \right), \tag{4}$$
$$\boldsymbol{v}_t = (1 - \alpha_t)\boldsymbol{v}_{t-1} + \alpha_t \mathscr{A}(\boldsymbol{x}_t; \xi_t) - \eta_{1,t}\boldsymbol{\delta}_t + \eta_{2,t}\boldsymbol{e}_t, \quad \boldsymbol{x}_{t+1} = \boldsymbol{x}_t - \gamma\boldsymbol{v}_t. \tag{5}$$

Here $(1 - \alpha_t)\boldsymbol{v}_{t-1} + \alpha_t\mathscr{A}(\boldsymbol{x}_t; \xi_t)$ originates from the original uncompressed algorithm. Other terms that depends on the compression error $\boldsymbol{\delta}_t$ indicates the influence of compression (see our supplementary material for more details about the parameter setting for different compression algorithms.)

## 4.1 Convergence Rate of General Algorithms

In this section, we decompose the convergence rate of ErrorCompensatedX into two parts: $\mathcal{R}_{\text{uncompressed}}(T)$ that only depends on the original algorithm without compression, and $\mathcal{R}_\epsilon$ which only depends on the way of error compensation.

Instead of investigating $\boldsymbol{x}_t$ directly, we introduce an auxiliary sequence $\{\hat{\boldsymbol{x}}_t\}$, which is defined as

$$\boldsymbol{u}_t = (1 - \alpha_t)\boldsymbol{u}_{t-1} + \alpha_t\mathscr{A}(\boldsymbol{x}_t; \xi_t), \quad \hat{\boldsymbol{x}}_t = \hat{\boldsymbol{x}}_{t-1} - \gamma\boldsymbol{u}_t, \tag{6}$$

and set $\hat{\boldsymbol{x}}_0 = \boldsymbol{x}_0$, $\boldsymbol{u}_0 = \boldsymbol{v}_0$ for initialization. Below we are going to see that $\{\hat{\boldsymbol{x}}_t\}$ admits some nice properties and is very helpful for the analysis of the convergence rate for $\{\boldsymbol{x}_t\}$.

For $\hat{\boldsymbol{x}}_t$, there seems to be no compression error, but we still compute the gradient estimator using $\boldsymbol{x}_t$ instead $\hat{\boldsymbol{x}}_t$ directly. Therefore we also define

$$\hat{\boldsymbol{u}}_0 = \mathscr{A}(\boldsymbol{x}_0; \xi_0) = \mathscr{A}(\hat{\boldsymbol{x}}_0; \xi_0), \quad \hat{\boldsymbol{u}}_t = (1 - \alpha_t)\hat{\boldsymbol{u}}_{t-1} + \alpha_t\mathscr{A}(\hat{\boldsymbol{x}}_t; \xi_t). \tag{7}$$

In this case, in the view of $\{\hat{\boldsymbol{x}}_t\}$, it basically updates the model following

$$\hat{\boldsymbol{u}}_t = (1 - \alpha_t)\hat{\boldsymbol{u}}_{t-1} + \alpha_t\mathscr{A}(\hat{\boldsymbol{x}}_t; \xi_t), \quad \hat{\boldsymbol{x}}_t = \hat{\boldsymbol{x}}_{t-1} - \gamma\hat{\boldsymbol{u}}_t - \underbrace{\gamma(\boldsymbol{u}_t - \hat{\boldsymbol{u}}_t)}_{\textbf{compression bias}}.$$

Therefore $\{\hat{\boldsymbol{x}}_t\}$ is essentially using the same uncompressed gradient estimator $\mathscr{A}(\hat{\boldsymbol{x}}_t; \xi_t)$ for constructing the gradient estimator $\hat{\boldsymbol{u}}_t$ except that when it updates the model, it will use $\hat{\boldsymbol{u}}_t$ plus a compression bias term $\gamma(\boldsymbol{u}_t - \hat{\boldsymbol{u}}_t)$.

Now we present the key theorem that concludes the convergence rate of any algorithm (for non-convex loss function) that updates the model according to (4) and (5). Before we introduce the final theorem, we first make some commonly used assumptions:

**Assumption 1.** *We assume $f^* > -\infty$ and make the following assumptions:*

1. ***Lipschitzian gradient:*** *$f(\cdot)$ and $F(\cdot; \xi)$ are assumed to be with $L$-smooth, which means*

$$\|\nabla F(\boldsymbol{x}; \xi) - \nabla F(\boldsymbol{y}; \xi)\| \leq L_F\|\boldsymbol{x} - \boldsymbol{y}\|, \quad \forall \boldsymbol{x}, \boldsymbol{y}, \xi,$$
$$\|\nabla f_i(\boldsymbol{x}) - \nabla f_i(\boldsymbol{y})\| \leq L\|\boldsymbol{x} - \boldsymbol{y}\|, \quad \forall \boldsymbol{x}, \boldsymbol{y}, i;$$

2. ***Bounded variance:*** *The variance of the stochastic gradient is bounded*

$$\mathbb{E}_{\xi^{(i)} \sim \mathcal{D}_i}\|\nabla F(\boldsymbol{x}; \xi^{(i)}) - \nabla f(\boldsymbol{x})\|^2 \leq \sigma^2, \quad \forall \boldsymbol{x}, i.$$

**Theorem 1.** *For algorithms that follow the updating rule* (4) *and* (5), *under Assumption 1, we have*

$$\frac{1}{T}\sum_{t=0}^{T-1}\mathbb{E}\|\nabla f(\boldsymbol{x}_t)\|^2 \leq \underbrace{\frac{16(f(\boldsymbol{x}_0)-f^*)}{\gamma T} + \frac{8}{T}\sum_{t=0}^{T-1}A_t}_{\mathcal{R}_{uncompressed}(T)} + \underbrace{\frac{64L_{\mathscr{A}}^2+2L^2}{T}\sum_{t=0}^{T-1}\mathbb{E}\|\boldsymbol{x}_t-\hat{\boldsymbol{x}}_t\|^2}_{\mathcal{R}_\epsilon},$$

*where $A_t$ is defined as*

$$A_t := \mathbb{E}\|\nabla f(\hat{\boldsymbol{x}}_t)-\hat{\boldsymbol{u}}_t\|^2 - (1-2L\gamma)\mathbb{E}\|\hat{\boldsymbol{u}}_t\|^2 - \frac{\mathbb{E}\|\nabla f(\hat{\boldsymbol{x}}_t)\|^2}{4}$$

*with $\hat{\boldsymbol{u}}_t$ defined in* (7), *and $L_{\mathscr{A}}$ is the Lipschitz constant of $\mathscr{A}(\cdot;\xi)$, i.e.,*

$$\|\mathscr{A}(\boldsymbol{x};\xi)-\mathscr{A}(\boldsymbol{y};\xi)\| \leq L_{\mathscr{A}}\|\boldsymbol{x}-\boldsymbol{y}\|, \quad \forall \boldsymbol{x},\boldsymbol{y},\xi.$$

Here $A_t$, which depends only on the original uncompressed algorithm, indicates the bias of constructing the gradient estimator using uncompressed $\mathscr{A}(\hat{\boldsymbol{x}}_t;\xi_t)$, hence $\frac{16(f(\boldsymbol{x}_0)-f^*)}{\gamma T} + \frac{8}{T}\sum_{t=0}^{T}A_t$ is usually the convergence rate of the original algorithm without compression.

In order to get a clear vision about the influence of the compression error under different error compensation methods, in the theorem below we are going to give an upper bound of $\sum_{t=0}^{T}\mathbb{E}\|\boldsymbol{x}_t-\hat{\boldsymbol{x}}_t\|^2$ for different error compensation methods.

**Assumption 2.** *The magnitude of the compression error are assumed to be bounded by a constant $\epsilon$:*

$$\mathbb{E}_\omega\left\|\boldsymbol{\delta}_t^{(i)}\right\|^2 \leq \frac{\epsilon^2}{2}, \quad \mathbb{E}_\omega\left\|\boldsymbol{\delta}_t^{(i)}\right\|^2 \leq \frac{\epsilon^2}{2}, \forall t,i.$$

**Theorem 2.** *For algorithms that follow the updating rule* (4) *and* (5), *under Assumption 2, setting $\beta=1$, $\eta_{1,t}=\eta_1$, $\eta_{2,t}=\eta_2$, $c_{1,t}=c_1$, and $c_{2,t}=c_2$, we have*

$$\boldsymbol{x}_t - \hat{\boldsymbol{x}}_t = -\frac{\eta_1}{\alpha}\sum_{s=0}^{t}\left(1-(1-\alpha)^{t-s+1}\right)\boldsymbol{\delta}_s + \frac{\eta_2 c_1}{\alpha}\sum_{s=0}^{t-1}\left(1-(1-\alpha)^{t-s}\right)\boldsymbol{\delta}_s$$

$$+ \frac{\eta_2 c_2}{\alpha}\sum_{s=0}^{t-2}\left(1-(1-\alpha)^{t-s-1}\right)\boldsymbol{\delta}_s.$$

*More specifically, we have*

- *No Compensation:* $\sum_{t=0}^{T}\mathbb{E}\|\boldsymbol{x}_t-\hat{\boldsymbol{x}}_t\|^2 \leq \frac{\gamma^2 T^2\epsilon^2}{\alpha^2}$;

- *Single Compensation:* $\sum_{t=0}^{T}\mathbb{E}\|\boldsymbol{x}_t-\hat{\boldsymbol{x}}_t\|^2 \leq \frac{\gamma^2\epsilon^2}{\alpha^2}$;

- *ErrorCompensatedX:* $\sum_{t=0}^{T}\mathbb{E}\|\boldsymbol{x}_t-\hat{\boldsymbol{x}}_t\|^2 \leq \gamma^2\alpha^2\epsilon^2.$

## 4.2 Convergence Rate for Different Gradient Estimators

In this section, we apply Theorem 1 to get the specific convergence rate of ErrorCompensatedX for Momentum SGD, STORM, and IGT.

### 4.2.1 SGD and Momentum SGD

Since SGD is a special case of Momentum SGD when setting $\alpha=1$, the following theorem includes both SGD and Momentum SGD.

**Theorem 3.** *Setting $\mathscr{A}(\boldsymbol{x}_t;\xi_t)=\nabla F(\boldsymbol{x}_t;\xi_t)$, if $L\gamma \leq \frac{\alpha}{12}$, under Assumptions 1 and 2, we have*

$$\sum_{t=0}^{T-1}A_t \leq \frac{17\gamma L\sigma^2 T}{3n}.$$

This leads us to the corollary below:

**Corollary 4.** *For ErrorCompensatedX, under Assumptions 1 and 2, setting $\alpha_t = \alpha \in (0,1]$, $\mathscr{A}(\boldsymbol{x}_t; \xi_t) = \nabla F(\boldsymbol{x}_t; \xi_t)$ and $\gamma = \min\left\{\frac{\alpha}{12L}, \sqrt{\frac{n}{T\sigma^2}}, \left(\frac{1}{\epsilon^2 T}\right)^{1/3}\right\}$, we have*

$$\frac{1}{T}\sum_{t=0}^{T-1} \mathbb{E}\|\nabla f(\boldsymbol{x}_t)\|^2 \lesssim \frac{\sigma}{\sqrt{nT}} + \frac{\alpha^2}{(\epsilon T)^{\frac{2}{3}}} + \frac{1}{\alpha T}.$$

The leading term admits the order of $\mathcal{O}\left(1/\sqrt{nT}\right)$, which is the same as uncompressed training.

### 4.2.2 STOchastic Recursive Momentum (STORM)

**Theorem 5.** *Setting $\mathscr{A}(\boldsymbol{x}_t; \xi_t) = \frac{1}{\alpha_t}\left(\nabla F(\boldsymbol{x}_t; \xi_t) - (1-\alpha_t)\nabla F(\boldsymbol{x}_{t-1}; \xi_t)\right)$, if $\gamma \leq \frac{1}{4L}$ and $\alpha \geq \frac{8(L^2+L_F^2)\gamma^2}{n}$, under Assumptions 1 and 2, we have*

$$\sum_{t=0}^{T-1} A_t \leq \frac{2\alpha\sigma^2 T}{n} + \frac{\sigma^2}{n\alpha B_0}.$$

This leads us to the corollary below:

**Corollary 6.** *For ErrorCompensatedX, under Assumptions 1 and 2, setting $\gamma = \min\left\{\frac{1}{4L}, \left(\frac{n^2}{\sigma^2 T}\right)^{\frac{1}{3}}, \left(\frac{n}{\epsilon^2 T}\right)^{\frac{1}{7}}\right\}$, $\alpha_t = \frac{8L^2\gamma^2}{n}$, $B_0 = \frac{\sigma^{\frac{8}{3}}T^{\frac{1}{3}}}{n^{\frac{2}{3}}}$, and $\mathscr{A}(\boldsymbol{x}_t; \xi_t) = \frac{1}{\alpha_t}\left(\nabla F(\boldsymbol{x}_t; \xi_t) - (1-\alpha_t)\nabla F(\boldsymbol{x}_{t-1}; \xi_t)\right)$, we have*

$$\frac{1}{T}\sum_{t=0}^{T-1} \mathbb{E}\|\nabla f(\boldsymbol{x}_t)\|^2 \lesssim \left(\frac{\sigma}{nT}\right)^{\frac{2}{3}} + \left(\frac{\epsilon^2}{nT^6}\right)^{\frac{1}{7}} + \frac{1}{T}.$$

The leading term admits the order of $\mathcal{O}\left(1/(nT)^{\frac{2}{3}}\right)$, which is the same as uncompressed training (Yuan et al., 2020).

### 4.2.3 Implicit Gradient Transport (IGT)

For IGT, we need to make some extra assumptions (as listed below) for theoretical analysis.

**Assumption 3.** *The Hessian $\nabla^2 f(\cdot)$ is $\rho$-Lipschitz continuous, i.e., for every $\boldsymbol{x}, \boldsymbol{y} \in \mathbb{R}^d$,*

$$\left\|\nabla^2 f(\boldsymbol{x}) - \nabla^2 f(\boldsymbol{y})\right\| \leq \rho\|\boldsymbol{x}-\boldsymbol{y}\|, \quad \forall \boldsymbol{x}, \boldsymbol{y}.$$

*The magnitude of the stochastic gradient is upper bounded, i.e.,*

$$\mathbb{E}\|\nabla F(\boldsymbol{x}; \xi) - \nabla F(\boldsymbol{y}; \xi)\|^2 \leq \Delta^2, \quad \forall \boldsymbol{x}, \boldsymbol{y}.$$

Now we are ready to present the result of IGT:

**Theorem 7.** *Setting $\mathscr{A}(\boldsymbol{x}_t; \xi_t) = \nabla F\left(\boldsymbol{x}_t + \frac{1-\alpha_t}{\alpha_t}(\boldsymbol{x}_t - \boldsymbol{x}_{t-1}); \xi_t\right)$, if $\gamma \leq \frac{1}{2L}$, under Assumptions 1, 2 and 3, we have*

$$\sum_{t=0}^{T-1} A_t \leq \frac{\alpha\sigma^2 T}{n} + \frac{\sigma^2}{\alpha n B_0} + \frac{\rho^2\gamma^4\Delta^4 T}{\alpha^4}.$$

This leads us to the corollary below:

**Corollary 8.** *For ErrorCompensatedX, under Assumptions 1, 2 and 3, by setting $\gamma = \min\left\{\frac{1}{2L}, \left(\frac{n^4}{\sigma^8 T^5}\right)^{\frac{1}{9}}, \left(\frac{n}{\epsilon^2 T}\right)^{\frac{1}{7}}\right\}$, $\alpha = \left(\frac{n^5}{\sigma^8 T^4}\right)^{\frac{1}{9}}$ and $B_0 = 1$, we have*

$$\frac{1}{T}\sum_{t=0}^{T-1} \mathbb{E}\|\nabla f(\boldsymbol{x}_t)\|^2 \lesssim \left(\frac{\sigma^8}{n^4 T^4}\right)^{\frac{1}{9}} + \left(\frac{\epsilon^2}{nT^6}\right)^{\frac{1}{7}} + \frac{1}{T}.$$

The leading term admits the order of $\mathcal{O}(1/(nT)^{\frac{4}{9}})$, which is even worse than that of Momentum SGD. A sharper rate of IGT for a general non-convex loss function is still an open problem.

# 5 Numerical Experiments

In this section, we train ResNet-50 (He et al., 2016) on CIFAR10, which consists of 50000 training images and 10000 testing images, each has 10 labels. We run the experiments on eight workers, each having a 1080Ti GPU. The batch size on each worker is 16 and the total batch size is 128. For each choice of $\mathscr{A}(\boldsymbol{x}_t; \xi_t)$, we evaluate four implementations: 1) **Original algorithm** without compression. 2) **No compensation**, which compresses the data directly without the previous information. 3) **Single compensation**, which compresses the gradient estimator with compression error from the last step added. 4) **ErrorCompensatedX**.

We use the $1$-bit compression in Tang et al. (2019), which leads to an overall $96\%$ of communication volume reduction. We find that for both STORM and IGT, setting $\alpha_t = \frac{1}{1+c_0 t}$, where $c_0$ is a constant and $t$ is the training step count, would make the training faster. We grid search the best learning rate from $\{0.5, 0.1, 0.001\}$ and $c_0$ from $\{0.1, 0.05, 0.001\}$, and find that the best learning rate is $0.01$ with $c_0 = 0.05$ for both original STORM and IGT. So we use this configuration for the other three implementations. For Momentum, usually we will not set $\alpha_t$ to be too small, therefore we only set $\alpha_t = 0.1$, and the best learning rate is $0.1$ after the same grid search. We set $\beta = 0.3$ for the low-pass filter in all cases.

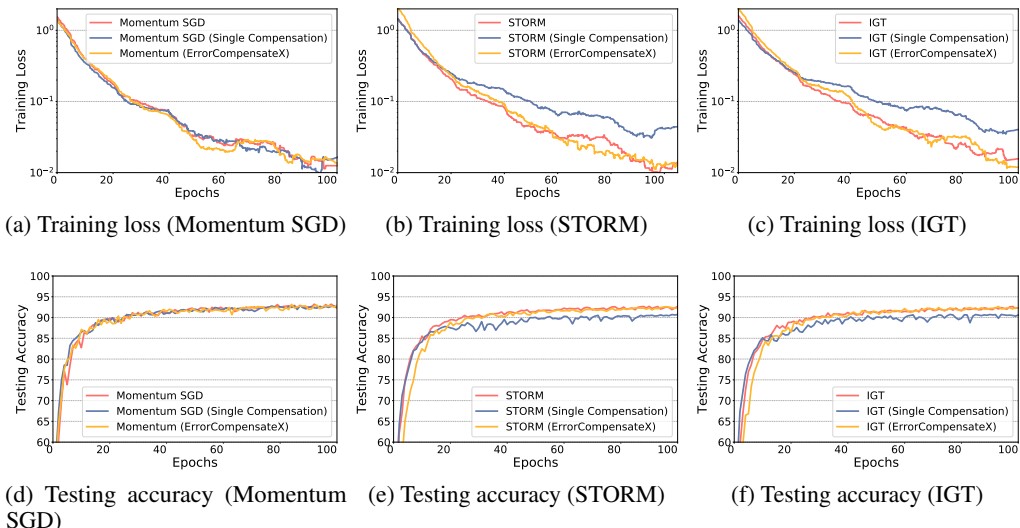

(a) Training loss (Momentum SGD)  (b) Training loss (STORM)  (c) Training loss (IGT)

(d) Testing accuracy (Momentum SGD)  (e) Testing accuracy (STORM)  (f) Testing accuracy (IGT)

Figure 2: Epoch-wise convergence comparison on ResNet-50 for Momenum SGD (left column), STORM (middle column), and IGT (right column) with different communication implementations. We do not include the result of no compensation because it diverges after a few steps of training.

As illustrated in Figure 2, for both STORM and IGT, ErrorCompensatedX achieves almost the same convergence speed with the original algorithm, while Single Compensation fails to achieve this. We find that with increasing magnitude of $c_0$, this gap of convergence speed would increase, which further validate our claim that ErrorCompensatedX is more essential when $\alpha_t$ gets small. For Momentum SGD, all the three implementations (except compression without error compensation) achieves similar convergence speed. This is because $\alpha_t = 0.1$ is comparably large.

# 6 Conclusion and Remarks

In this paper, we address an important problem for communication efficient distributed training: how to fully compensate the compression error for variance reduced optimization algorithms. In our paper, we consider a more general class of optimization algorithms (including SGD and Momentum SGD), and we propose a novel method: ErrorCompensatedX, which utilize the compression error from the last two steps, in order to fully compensate the history compression error while previous method fails to do. From the theoretical perspective, we provide a unified theoretical analysis framework that gives an intuitive evaluation for the side-effect of the compression, and shows that

ErrorCompensatedX admits the same asymptotic convergence rate with each of the original algorithm. Numerical experiments are implemented to show ErrorCompensatedX's convergence and its better performance comparing to other implementations.

## Acknowledgments

Yao Li and Ming Yan are partially supported by the NSF grant DMS-2012439 and a Facebook Faculty Research Award.

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
