# Supplementary

In the supplementary, we first prove the updating rule of ErrorCompensatedX. Then we will present the detailed proof for Theorem 1 followed by the detailed convergence rate result for each algorithm.

## 7  Updating rule of ErrorCompensatedX

The updating rule of Algorithm 1 admits the form

$$
\overline{\boldsymbol{e}}_t = (1 - \beta)\overline{\boldsymbol{e}}_{t-1} + \beta \left( \frac{\alpha_{t-1}}{\alpha_t}(2 - \alpha_t)\overline{\boldsymbol{\delta}}_{t-1} + \frac{\alpha_{t-2}}{\alpha_t}(1 - \alpha_t)\overline{\boldsymbol{\delta}}_{t-2} \right),
$$

$$
\boldsymbol{v}_t = (1 - \alpha_t)\boldsymbol{v}_{t-1} + \alpha_t \overline{\boldsymbol{b}}_t(\boldsymbol{x}_t) - \alpha_t \overline{\boldsymbol{\delta}}_t + \alpha_t \overline{\boldsymbol{e}}_t,
$$

$$
\boldsymbol{x}_{t+1} = \boldsymbol{x}_t - \gamma \boldsymbol{v}_t.
$$

where

$$
\overline{\boldsymbol{e}}_t := \frac{1}{n}\sum_i \boldsymbol{e}_t^{(i)} + \boldsymbol{e}_t,
$$

$$
\overline{\boldsymbol{\delta}}_t := \boldsymbol{\delta}_t + \frac{1}{n}\sum_{i=1}^n \boldsymbol{\delta}_t^{(i)},
$$

$$
\overline{\boldsymbol{b}}_t(\boldsymbol{x}) := \frac{1}{n}\sum_i \mathscr{A}\left(\boldsymbol{x}_t; \xi_t^{(i)}\right).
$$

*Proof.* From the algorithm description in Algorithm 1, we shall see that we are essentially using $\boldsymbol{v}_t$ as the gradient estimator of the full gradient. From the updating rule of $\boldsymbol{v}_t$, we have

$$
\begin{aligned}
\boldsymbol{v}_t &= (1 - \alpha_t)\boldsymbol{v}_{t-1} + \alpha_t C_\omega\left[\Delta_t\right]\\
&= (1 - \alpha_t)\boldsymbol{v}_{t-1} + \alpha_t \left(\Delta_t - \boldsymbol{\delta}_t\right)\\
&= (1 - \alpha_t)\boldsymbol{v}_{t-1} + \frac{\alpha_t}{n}\sum_i C_\omega\left[\Delta_t^{(i)}\right] + \alpha_t \boldsymbol{e}_t - \alpha_t \boldsymbol{\delta}_t\\
&= (1 - \alpha_t)\boldsymbol{v}_{t-1} + \frac{\alpha_t}{n}\sum_i \left(\Delta_t^{(i)} - \boldsymbol{\delta}_t^{(i)}\right) + \alpha_t \boldsymbol{e}_t - \alpha_t \boldsymbol{\delta}_t\\
&= (1 - \alpha_t)\boldsymbol{v}_{t-1} + \frac{\alpha_t}{n}\sum_i \left(\mathscr{A}\left(\boldsymbol{x}_t; \xi_t^{(i)}\right) + \boldsymbol{e}_t^{(i)} - \boldsymbol{\delta}_t^{(i)}\right) + \alpha_t \boldsymbol{e}_t - \alpha_t \boldsymbol{\delta}_t\\
&= (1 - \alpha)\boldsymbol{v}_{t-1} + \alpha_t \overline{\boldsymbol{b}}_t(\boldsymbol{x}_t) + \alpha_t \overline{\boldsymbol{e}}_t - \alpha_t \overline{\boldsymbol{\delta}}_t.
\end{aligned}
$$

Notice that in the deduction above we continuously using the fact that $C_\omega[\boldsymbol{x}] = \boldsymbol{x} - \boldsymbol{\delta}$. For $\overline{\boldsymbol{e}}_t$, we have

$$
\begin{aligned}
\overline{\boldsymbol{e}}_t &= \frac{1}{n}\sum_i \boldsymbol{e}_t^{(i)} + \boldsymbol{e}_t\\
&= \frac{1 - \beta}{n}\sum_i \boldsymbol{e}_{t-1}^{(i)} + \frac{\beta}{n}\sum_i \left(\frac{\alpha_{t-1}}{\alpha_t}(2 - \alpha_t)\delta_{t-1}^{(i)} + \frac{\alpha_{t-2}}{\alpha_t}(1 - \alpha_t)\delta_{t-2}^{(i)}\right)\\
&\quad + (1 - \beta)\boldsymbol{e}_{t-1} + \beta\left(\frac{\alpha_{t-1}}{\alpha_t}(2 - \alpha_t)\delta_{t-1} + \frac{\alpha_{t-2}}{\alpha_t}(1 - \alpha_t)\delta_{t-2}\right)\\
&= (1 - \beta)\overline{\boldsymbol{e}}_{t-1} + \beta\left(\frac{\alpha_{t-1}}{\alpha_t}(2 - \alpha_t)\overline{\boldsymbol{\delta}}_{t-1} + \frac{\alpha_{t-2}}{\alpha_t}(1 - \alpha_t)\overline{\boldsymbol{\delta}}_{t-2}\right),
\end{aligned}
$$

which completes the proof. $\qquad\square$

## 8  General Formulation of the Updating Rule

As mentioned in Section 4, all the error compensation mentioned so far admits the updating rule of (4) and (5). Therefore in Table 2 we list the specific choice of parameters for each algorithm.

|  | $\eta_{1,t}$ | $\eta_{2,t}$ | $c_{1,t}$ | $c_{2,t}$ |
|---|---|---|---|---|
| **Without Compensation** | 1 | 0 | 0 | 0 |
| **Single Compensation** | 1 | 1 | 1 | 0 |
| **ErrorCompensatedX** | $\alpha_t$ | $\alpha_t$ | $2 - \alpha_t$ | $1 - \alpha_t$ |

Table 2: Different choices of $\eta_{1,t}$, $\eta_{2,t}$, $c_{1,t}$, and $c_{2,t}$ for each algorithm.

# 9 Proof of Theorems

## 9.1 Proof of Theorem 1

*Proof.* In the view of $\{\hat{\boldsymbol{x}}_t\}$, we have

$$
\begin{aligned}
&\mathbb{E}f(\hat{\boldsymbol{x}}_{t+1}) - \mathbb{E}f(\hat{\boldsymbol{x}}_t) \\
\leq& \mathbb{E}\langle \nabla f(\hat{\boldsymbol{x}}_t), \hat{\boldsymbol{x}}_{t+1} - \hat{\boldsymbol{x}}_t \rangle + \frac{L}{2}\mathbb{E}\|\hat{\boldsymbol{x}}_{t+1} - \hat{\boldsymbol{x}}_t\|^2 \\
=& -\gamma\mathbb{E}\langle \nabla f(\hat{\boldsymbol{x}}_t), \boldsymbol{u}_t \rangle + \frac{L\gamma^2}{2}\mathbb{E}\|\boldsymbol{u}_t\|^2 \\
=& -\gamma\mathbb{E}\langle \nabla f(\hat{\boldsymbol{x}}_t), \hat{\boldsymbol{u}}_t + (\boldsymbol{u}_t - \hat{\boldsymbol{u}}_t) \rangle + \frac{L\gamma^2}{2}\mathbb{E}\|\hat{\boldsymbol{u}}_t + (\boldsymbol{u}_t - \hat{\boldsymbol{u}}_t)\|^2 \\
\leq& -\gamma\mathbb{E}\langle \nabla f(\hat{\boldsymbol{x}}_t), \hat{\boldsymbol{u}}_t \rangle - \gamma\mathbb{E}\langle \nabla f(\hat{\boldsymbol{x}}_t), \boldsymbol{u}_t - \hat{\boldsymbol{u}}_t \rangle + L\gamma^2\mathbb{E}\|\hat{\boldsymbol{u}}_t\|^2 + L\gamma^2\mathbb{E}\|\boldsymbol{u}_t - \hat{\boldsymbol{u}}_t\|^2 \\
\leq& -\gamma\mathbb{E}\langle \nabla f(\hat{\boldsymbol{x}}_t), \hat{\boldsymbol{u}}_t \rangle + \frac{\gamma}{4}\mathbb{E}\|\nabla f(\hat{\boldsymbol{x}}_t)\|^2 + \gamma\mathbb{E}\|\boldsymbol{u}_t - \hat{\boldsymbol{u}}_t\|^2 + L\gamma^2\mathbb{E}\|\hat{\boldsymbol{u}}_t\|^2 + L\gamma^2\mathbb{E}\|\boldsymbol{u}_t - \hat{\boldsymbol{u}}_t\|^2 \\
\leq& -\frac{\gamma}{2}\left(\mathbb{E}\|\nabla f(\hat{\boldsymbol{x}}_t)\|^2 + \mathbb{E}\|\hat{\boldsymbol{u}}_t\|^2 - \mathbb{E}\|\nabla f(\hat{\boldsymbol{x}}_t) - \hat{\boldsymbol{u}}_t\|^2\right) + \frac{\gamma}{4}\mathbb{E}\|\nabla f(\hat{\boldsymbol{x}}_t)\|^2 \\
&+ \gamma\|\boldsymbol{u}_t - \hat{\boldsymbol{u}}_t\|^2 + L\gamma^2\mathbb{E}\|\hat{\boldsymbol{u}}_t\|^2 + L\gamma^2\mathbb{E}\|\boldsymbol{u}_t - \hat{\boldsymbol{u}}_t\|^2 \\
\leq& -\frac{\gamma}{8}\mathbb{E}\|\nabla f(\hat{\boldsymbol{x}}_t)\|^2 - \frac{\gamma}{2}\left((1 - 2L\gamma)\mathbb{E}\|\hat{\boldsymbol{u}}_t\|^2 - \mathbb{E}\|\nabla f(\hat{\boldsymbol{x}}_t) - \hat{\boldsymbol{u}}_t\|^2 + \frac{\mathbb{E}\|\nabla f(\hat{\boldsymbol{x}}_t)\|^2}{4}\right) \\
&+ (1 + L\gamma)\gamma\mathbb{E}\|\boldsymbol{u}_t - \hat{\boldsymbol{u}}_t\|^2.
\end{aligned}
$$

Summing up the inequality above from $t = 0$ to $t = T$, with rearrangement we get

$$
\begin{aligned}
&\frac{1}{T}\sum_{t=0}^{T-1}\mathbb{E}\|\nabla f(\hat{\boldsymbol{x}}_t)\|^2 \\
\leq& \frac{8(f(\hat{\boldsymbol{x}}_0) - f(\hat{\boldsymbol{x}}_T))}{\gamma T} - \frac{4}{T}\sum_{t=0}^{T-1}\left((1 - 2L\gamma)\mathbb{E}\|\hat{\boldsymbol{u}}_t\|^2 - \mathbb{E}\|\nabla f(\hat{\boldsymbol{x}}_t) - \hat{\boldsymbol{u}}_t\|^2 + \frac{\mathbb{E}\|\nabla f(\hat{\boldsymbol{x}}_t)\|^2}{4}\right) \\
&+ \frac{8 + 8L\gamma}{T}\sum_{t=0}^{T-1}\mathbb{E}\|\boldsymbol{u}_t - \hat{\boldsymbol{u}}_t\|^2 \\
\leq& \frac{8(f(\boldsymbol{x}_0) - f^*)}{\gamma T} - \frac{4}{T}\sum_{t=0}^{T-1}\left((1 - 2L\gamma)\mathbb{E}\|\hat{\boldsymbol{u}}_t\|^2 - \mathbb{E}\|\nabla f(\hat{\boldsymbol{x}}_t) - \hat{\boldsymbol{u}}_t\|^2 + \frac{\mathbb{E}\|\nabla f(\hat{\boldsymbol{x}}_t)\|^2}{4}\right) \\
&+ \frac{8 + 8L\gamma}{T}\sum_{t=0}^{T-1}\mathbb{E}\|\boldsymbol{u}_t - \hat{\boldsymbol{u}}_t\|^2. \quad\quad (8)
\end{aligned}
$$

In order to upper bound $\mathbb{E}\|\boldsymbol{u}_t - \hat{\boldsymbol{u}}_t\|^2$, we have

$$
\begin{aligned}
&\mathbb{E}\|\boldsymbol{u}_t - \hat{\boldsymbol{u}}_t\|^2 \\
=&\mathbb{E}\|(1-\alpha)(\boldsymbol{u}_{t-1} - \hat{\boldsymbol{u}}_{t-1}) + \alpha(\mathscr{A}(\boldsymbol{x}_t; \xi_t) - \mathscr{A}(\hat{\boldsymbol{x}}_t; \xi_t))\|^2 \\
\leq&\, (1+\alpha)\,\mathbb{E}\|(1-\alpha)(\boldsymbol{u}_{t-1} - \hat{\boldsymbol{u}}_{t-1})\|^2 + \left(1 + \frac{1}{\alpha}\right)\mathbb{E}\|\alpha(\mathscr{A}(\boldsymbol{x}_t; \xi_t) - \mathscr{A}(\hat{\boldsymbol{x}}_t; \xi_t))\|^2 \\
=&\,\left(1-\alpha^2\right)(1-\alpha)\mathbb{E}\|\boldsymbol{u}_{t-1} - \hat{\boldsymbol{u}}_{t-1}\|^2 + \left(\alpha^2 + \alpha\right)\mathbb{E}\|\mathscr{A}(\boldsymbol{x}_t; \xi_t) - \mathscr{A}(\hat{\boldsymbol{x}}_t; \xi_t)\|^2 \\
\leq&\,(1-\alpha)\,\mathbb{E}\|\boldsymbol{u}_{t-1} - \hat{\boldsymbol{u}}_{t-1}\|^2 + 2\alpha\mathbb{E}\|\mathscr{A}(\boldsymbol{x}_t; \xi_t) - \mathscr{A}(\hat{\boldsymbol{x}}_t; \xi_t)\|^2 \\
=&\,(1-\alpha)^t\mathbb{E}\|\boldsymbol{u}_0 - \boldsymbol{u}_0(\boldsymbol{y})\|^2 + 2\alpha \sum_{s=0}^{t}(1-\alpha)^{t-s}\mathbb{E}\|\mathscr{A}(\boldsymbol{x}_s; \xi_s) - \mathscr{A}(\hat{\boldsymbol{x}}_s; \xi_s)\|^2 \\
=&\,2\alpha \sum_{s=0}^{t}(1-\alpha)^{t-s}\mathbb{E}\|\mathscr{A}(\boldsymbol{x}_s; \xi_s) - \mathscr{A}(\hat{\boldsymbol{x}}_s; \xi_s)\|^2 \quad (\boldsymbol{u}_0 - \boldsymbol{u}_0(\boldsymbol{y}) = \mathbf{0}) \\
\leq&\,2\alpha L_{\mathscr{A}}^2 \sum_{s=0}^{t}(1-\alpha)^{t-s}\mathbb{E}\|\boldsymbol{x}_s - \hat{\boldsymbol{x}}_s\|^2.
\end{aligned}
$$

Therefore we have

$$
\begin{aligned}
\sum_{t=0}^{T-1}\mathbb{E}\|\boldsymbol{u}_t - \hat{\boldsymbol{u}}_t\|^2 \leq&\,2\alpha L_{\mathscr{A}}^2 \sum_{t=0}^{T-1}\sum_{s=0}^{t}(1-\alpha)^{t-s}\mathbb{E}\|\boldsymbol{x}_s - \hat{\boldsymbol{x}}_s\|^2 \\
=&\,2\alpha L_{\mathscr{A}}^2 \sum_{s=0}^{T-1}\sum_{t=s}^{T-1}(1-\alpha)^{t-s}\mathbb{E}\|\boldsymbol{x}_s - \hat{\boldsymbol{x}}_s\|^2 \\
\leq&\,2L_{\mathscr{A}}^2 \sum_{t=0}^{T-1}\mathbb{E}\|\boldsymbol{x}_t - \hat{\boldsymbol{x}}_t\|^2.
\end{aligned}
\tag{9}
$$

Combining (8) and (9) together we get

$$
\begin{aligned}
&\frac{1}{T}\sum_{t=0}^{T}\mathbb{E}\|\nabla f(\hat{\boldsymbol{x}}_t)\|^2 \\
\leq&\frac{8(f(\boldsymbol{x}_0) - f^*)}{\gamma T} - \frac{4}{T}\sum_{t=0}^{T}\left((1-2L\gamma)\mathbb{E}\|\hat{\boldsymbol{u}}_t\|^2 - \mathbb{E}\|\nabla f(\hat{\boldsymbol{x}}_t) - \hat{\boldsymbol{u}}_t\|^2 + \frac{\mathbb{E}\|\nabla f(\hat{\boldsymbol{x}}_t)\|^2}{4}\right) \\
&+ \frac{16(1+L\gamma)L_{\mathscr{A}}^2}{T}\sum_{t=0}^{T}\mathbb{E}\|\boldsymbol{x}_t - \hat{\boldsymbol{x}}_t\|^2.
\end{aligned}
$$

Hence $\|\nabla f(\boldsymbol{x}_t)\|^2$ can be upper bounded by using

$$\frac{1}{T}\sum_{t=0}^{T-1}\mathbb{E}\|\nabla f(\boldsymbol{x}_t)\|^2$$

$$\leq\frac{1}{T}\sum_{t=0}^{T-1}\mathbb{E}\|\nabla f(\hat{\boldsymbol{x}}_t) + (\nabla f(\boldsymbol{x}_t) - \nabla f(\hat{\boldsymbol{x}}_t))\|^2$$

$$\leq\frac{2}{T}\sum_{t=0}^{T-1}\mathbb{E}\|\nabla f(\hat{\boldsymbol{x}}_t)\|^2 + \frac{2}{T}\sum_{t=0}^{T-1}\mathbb{E}\|\nabla f(\boldsymbol{x}_t) - \nabla f(\hat{\boldsymbol{x}}_t)\|^2$$

$$\leq\frac{2}{T}\sum_{t=0}^{T-1}\mathbb{E}\|\nabla f(\hat{\boldsymbol{x}}_t)\|^2 + \frac{2L^2}{T}\sum_{t=0}^{T-1}\mathbb{E}\|\boldsymbol{x}_t - \hat{\boldsymbol{x}}_t\|^2$$

$$\leq\frac{16(f(\boldsymbol{x}_0) - f^*)}{\gamma T} - \frac{8}{T}\sum_{t=0}^{T-1}\left((1 - 2L\gamma)\mathbb{E}\|\hat{\boldsymbol{u}}_t\|^2 - \mathbb{E}\|\nabla f(\hat{\boldsymbol{x}}_t) - \hat{\boldsymbol{u}}_t\|^2 + \frac{\mathbb{E}\|\nabla f(\hat{\boldsymbol{x}}_t)\|^2}{4}\right)$$

$$+ \frac{32(1 + L\gamma)L_{\mathscr{A}}^2 + 2L^2}{T}\sum_{t=0}^{T-1}\mathbb{E}\|\boldsymbol{u}_t - \hat{\boldsymbol{u}}_t\|^2.$$

If $L\gamma \leq 1$, we have

$$\frac{1}{T}\sum_{t=0}^{T-1}\mathbb{E}\|\nabla f(\boldsymbol{x}_t)\|^2$$

$$\leq\frac{16(f(\boldsymbol{x}_0) - f^*)}{\gamma T} - \frac{8}{T}\sum_{t=0}^{T-1}\left((1 - 2L\gamma)\mathbb{E}\|\hat{\boldsymbol{u}}_t\|^2 - \mathbb{E}\|\nabla f(\hat{\boldsymbol{x}}_t) - \hat{\boldsymbol{u}}_t\|^2 + \frac{\mathbb{E}\|\nabla f(\hat{\boldsymbol{x}}_t)\|^2}{4}\right)$$

$$+ \frac{64L_{\mathscr{A}}^2 + 2L^2}{T}\sum_{t=0}^{T-1}\mathbb{E}\|\boldsymbol{x}_t - \hat{\boldsymbol{x}}_t\|^2,$$

which completes the proof. $\qquad\square$

## 9.2 Proof of Theorem 3, 5 and 7

In this section, we are going to present the proof of theorems for different gradient estimators. We start with the key lemma for **Momentum SGD.**

**Lemma 9.** *For two nonnegative sequences $\{a_t\}$ and $\{b_t\}$ that satisfy*

$$a_t \leq \rho a_{t-1} + b_t,$$

*where $\rho \in (0, 1)$ is a constant, we have*

$$\sum_{t=0}^{T} a_t \leq \frac{\sum_{t=1}^{T} b_t + a_0}{1 - \rho}.$$

*Proof.* Since

$$a_t \leq \rho a_{t-1} + b_t,$$

adding the inequality above from $t = 1$ to $t = T$ we get

$$\sum_{t=1}^{T} a_t \leq \rho\sum_{t=0}^{T-1} a_t + \sum_{t=1}^{T} b_t$$

$$\leq \rho\sum_{t=0}^{T} a_t + \sum_{t=1}^{T} b_t.$$

Adding $a_0$ to both side we get

$$\sum_{t=0}^{T} a_t \leq \rho \sum_{t=0}^{T} a_t + \sum_{t=1}^{T} b_t + a_0$$

and the proof is complete by combining the sum of $a_t$ and dividing both sides by $1 - \rho$. $\qquad\square$

*Proof of Theorem 3.* For **Momentum SGD**, denote $\boldsymbol{m}_t = \alpha \sum_{s=0}^{t} (1-\alpha)^{t-s} \nabla f(\boldsymbol{x}_s)$, then we have

$$
\begin{aligned}
&\mathbb{E}\|\nabla f(\boldsymbol{x}_t) - \boldsymbol{u}_t\|^2 - \mathbb{E}\|\boldsymbol{u}_t\|^2 \\
=&\mathbb{E}\|\nabla f(\boldsymbol{x}_t)\|^2 - 2\mathbb{E}\langle \nabla f(\boldsymbol{x}_t), \boldsymbol{u}_t \rangle \\
=&\mathbb{E}\|\nabla f(\boldsymbol{x}_t)\|^2 - 2\mathbb{E}\langle \nabla f(\boldsymbol{x}_t), \boldsymbol{m}_t \rangle - 2\mathbb{E}\langle \nabla f(\boldsymbol{x}_t), \boldsymbol{u}_t - \boldsymbol{m}_t \rangle \\
=&-\mathbb{E}\|\boldsymbol{m}_t\|^2 + \mathbb{E}\|\nabla f(\boldsymbol{x}_t) - \boldsymbol{m}_t\|^2 - 2\mathbb{E}\langle \nabla f(\boldsymbol{x}_t), \boldsymbol{u}_t - \boldsymbol{m}_t \rangle.
\end{aligned}
\tag{10}
$$

For $\mathbb{E}\|\nabla f(\boldsymbol{x}_t) - \boldsymbol{m}_t\|^2$ we have

$$
\begin{aligned}
&\mathbb{E}\|\nabla f(\boldsymbol{x}_t) - \boldsymbol{m}_t\|^2 \\
=&\mathbb{E}\|\nabla f(\boldsymbol{x}_t) - (1-\alpha)\boldsymbol{m}_{t-1} - \alpha f(\boldsymbol{x}_t)\|^2 \\
=&\mathbb{E}\|(1-\alpha)(\nabla f(\boldsymbol{x}_{t-1}) - \boldsymbol{m}_{t-1} + \nabla f(\boldsymbol{x}_t) - \nabla f(\boldsymbol{x}_{t-1}))\|^2 \\
\leq&(1+\alpha)(1-\alpha)^2\mathbb{E}\|\nabla f(\boldsymbol{x}_{t-1}) - \boldsymbol{m}_{t-1}\|^2 + \left(1 + \frac{1}{\alpha}\right)(1-\alpha)^2\mathbb{E}\|\nabla f(\boldsymbol{x}_t) - \nabla f(\boldsymbol{x}_{t-1}))\|^2 \\
\leq&(1-\alpha)\mathbb{E}\|\nabla f(\boldsymbol{x}_{t-1}) - \boldsymbol{m}_{t-1}\|^2 + \frac{1}{\alpha}\mathbb{E}\|\nabla f(\boldsymbol{x}_t) - \nabla f(\boldsymbol{x}_{t-1}))\|^2 \\
\leq&(1-\alpha)\mathbb{E}\|\nabla f(\boldsymbol{x}_{t-1}) - \boldsymbol{m}_{t-1}\|^2 + \frac{L^2\gamma^2}{\alpha}\mathbb{E}\|\boldsymbol{u}_t\|^2.
\end{aligned}
$$

Therefore from Lemme 9,

$$\sum_{t=0}^{T-1} \mathbb{E}\|\nabla f(\boldsymbol{x}_t) - \boldsymbol{m}_t\|^2 \leq \frac{L^2\gamma^2}{\alpha^2} \sum_{t=0}^{T-1} \mathbb{E}\|\boldsymbol{u}_t\|^2. \tag{11}$$

For $\mathbb{E}\langle \nabla f(\boldsymbol{x}_t), \boldsymbol{u}_t - \boldsymbol{m}_t \rangle$, we have

$$
\begin{aligned}
\mathbb{E}\langle \nabla f(\boldsymbol{x}_t), \boldsymbol{u}_t - \boldsymbol{m}_t \rangle =&\mathbb{E}\langle \nabla f(\boldsymbol{x}_t), (1-\alpha)(\boldsymbol{u}_{t-1} - \boldsymbol{m}_{t-1}) + \alpha(\boldsymbol{g}_t - \nabla f(\boldsymbol{x}_t)) \rangle \\
=&\mathbb{E}\langle \nabla f(\boldsymbol{x}_t), (1-\alpha)(\boldsymbol{u}_{t-1} - \boldsymbol{m}_{t-1}) \rangle \\
=&(1-\alpha)\mathbb{E}\langle \nabla f(\boldsymbol{x}_{t-1}), \boldsymbol{u}_{t-1} - \boldsymbol{m}_{t-1} \rangle \\
&+ (1-\alpha)\mathbb{E}\langle \nabla f(\boldsymbol{x}_t) - \nabla f(\boldsymbol{x}_{t-1}), \boldsymbol{u}_{t-1} - \boldsymbol{m}_{t-1} \rangle,
\end{aligned}
\tag{12}
$$

where $\boldsymbol{g}_t = \nabla F(\boldsymbol{x}_t; \xi_t)$.

Notice that for $\nabla f(\boldsymbol{x}_t) - \nabla f(\boldsymbol{x}_{t-1})$, we have

$$
\begin{aligned}
&\mathbb{E}\|\nabla f(\boldsymbol{x}_t) - \nabla f(\boldsymbol{x}_{t-1})\|^2 \\
\leq&L^2\gamma^2\mathbb{E}\|\boldsymbol{u}_{t-1} - \boldsymbol{m}_{t-1} + \boldsymbol{m}_{t-1}\|^2 \\
\leq&2L^2\gamma^2\mathbb{E}\|\boldsymbol{m}_{t-1}\|^2 + 2L^2\gamma^2\mathbb{E}\|\boldsymbol{u}_{t-1} - \boldsymbol{m}_{t-1}\|^2.
\end{aligned}
\tag{13}
$$

Combining (12) and (13) together we get

$$
\begin{aligned}
&\mathbb{E}\langle \nabla f(\boldsymbol{x}_t), \boldsymbol{u}_t - \boldsymbol{m}_t \rangle \\
\leq&(1-\alpha)\mathbb{E}\langle \nabla f(\boldsymbol{x}_{t-1}), \boldsymbol{u}_{t-1} - \boldsymbol{m}_{t-1} \rangle + (1-\alpha)\mathbb{E}\left(\frac{1}{2\gamma L}\|\nabla f(\boldsymbol{x}_t) - \nabla f(\boldsymbol{x}_{t-1})\|^2 + \frac{\gamma L}{2}\mathbb{E}\|\boldsymbol{u}_{t-1} - \boldsymbol{m}_{t-1}\|^2\right) \\
\leq&(1-\alpha)\mathbb{E}\langle \nabla f(\boldsymbol{x}_{t-1}), \boldsymbol{u}_{t-1} - \boldsymbol{m}_{t-1} \rangle + (1-\alpha)\mathbb{E}\left(\gamma L\mathbb{E}\|\boldsymbol{m}_{t-1}\|^2 + \frac{3\gamma L}{2}\mathbb{E}\|\boldsymbol{u}_{t-1} - \boldsymbol{m}_{t-1}\|^2\right),
\end{aligned}
$$

Denote $c_t := \gamma L\mathbb{E}\|\boldsymbol{m}_{t-1}\|^2 + \frac{3\gamma L}{2}\mathbb{E}\|\boldsymbol{u}_{t-1} - \boldsymbol{m}_{t-1}\|^2$, then we get

$$\mathbb{E}\langle \nabla f(\boldsymbol{x}_t), \boldsymbol{u}_t - \mathbb{E}\boldsymbol{u}_t \rangle = (1-\alpha)^t\mathbb{E}\langle \nabla f(\boldsymbol{x}_0), \boldsymbol{u}_0 - \boldsymbol{m}_0 \rangle + \sum_{s=1}^{t}(1-\alpha)^{t-s}c_s.$$

Since $\mathbb{E}\mathscr{A}(\boldsymbol{x}_0;\xi_0) = \mathbb{E}\nabla F(\boldsymbol{x}_0;\xi_0) = \nabla F(\boldsymbol{x}_0)$, which means $\mathbb{E}\langle \nabla f(\boldsymbol{x}_0), \boldsymbol{u}_0 - \boldsymbol{m}_0\rangle = 0$. So the equation above becomes

$$\mathbb{E}\langle \nabla f(\boldsymbol{x}_t), \boldsymbol{u}_t - \boldsymbol{m}_t\rangle = \sum_{s=1}^{t}(1-\alpha)^{t-s}c_s,$$

and

$$\sum_{t=0}^{T-1}\mathbb{E}\langle \nabla f(\boldsymbol{x}_t), \boldsymbol{u}_t - \mathbb{E}\boldsymbol{u}_t\rangle \leq \frac{\sum_{t=0}^{T-1}|c_t|}{1-(1-\alpha)}$$

$$= \frac{\gamma L}{\alpha}\sum_{t=0}^{T-1}\mathbb{E}\|\boldsymbol{m}_t\|^2 + \frac{3\gamma L}{2\alpha}\sum_{t=0}^{T-1}\mathbb{E}\|\boldsymbol{u}_t - \boldsymbol{m}_t\|^2. \tag{14}$$

Combing (14), (11) and (10), we get

$$\sum_{t=0}^{T-1}\left(\mathbb{E}\|\nabla f(\boldsymbol{x}_t) - \boldsymbol{u}_t\|^2 - (1-2L\gamma)\mathbb{E}\|\boldsymbol{u}_t\|^2\right)$$

$$\leq -\sum_{t=0}^{T-1}\mathbb{E}\|\boldsymbol{m}_t\|^2 + \left(\frac{L^2\gamma^2}{\alpha^2} + 2L\gamma\right)\sum_{t=0}^{T-1}\mathbb{E}\|\boldsymbol{u}_t\|^2 + \frac{2\gamma L}{\alpha}\sum_{t=0}^{T-1}\mathbb{E}\|\boldsymbol{m}_t\|^2 + \frac{3\gamma L}{\alpha}\sum_{t=0}^{T-1}\mathbb{E}\|\boldsymbol{u}_t - \boldsymbol{m}_t\|^2$$

$$= -\left(1 - \frac{2L^2\gamma^2}{\alpha^2} - 4L\gamma - \frac{2\gamma L}{\alpha}\right)\sum_{t=0}^{T-1}\mathbb{E}\|\boldsymbol{m}_t\|^2 + \left(\frac{2L^2\gamma^2}{\alpha^2} + \frac{3\gamma L}{\alpha} + 4L\gamma\right)\sum_{t=0}^{T-1}\mathbb{E}\|\boldsymbol{u}_t - \boldsymbol{m}_t\|^2, \tag{15}$$

For $\mathbb{E}\|\boldsymbol{u}_t - \boldsymbol{m}_t\|^2$, we have

$$\mathbb{E}\|\boldsymbol{u}_t - \boldsymbol{m}_t\|^2 = \mathbb{E}\|(1-\alpha)(\boldsymbol{u}_{t-1} - \boldsymbol{m}_{t-1}) + \alpha(\boldsymbol{g}_t - \nabla f(\boldsymbol{x}_t))\|^2$$

$$= \mathbb{E}_{1:t-1}\mathbb{E}_t\|(1-\alpha)(\boldsymbol{u}_{t-1} - \boldsymbol{m}_{t-1}) + \alpha(\boldsymbol{g}_t - \nabla f(\boldsymbol{x}_t))\|^2$$

$$= \mathbb{E}_{1:t-1}\mathbb{E}_t\|(1-\alpha)(\boldsymbol{u}_{t-1} - \boldsymbol{m}_{t-1})\|^2 + \mathbb{E}_{1:t-1}\mathbb{E}_t\|\alpha(\boldsymbol{g}_t - \nabla f(\boldsymbol{x}_t))\|^2$$

$$\leq (1-\alpha)^2\mathbb{E}\|\boldsymbol{u}_{t-1} - \boldsymbol{m}_{t-1}\|^2 + \alpha^2\sigma^2$$

$$\leq (1-\alpha)\mathbb{E}\|\boldsymbol{u}_{t-1} - \boldsymbol{m}_{t-1}\|^2 + \alpha^2\sigma^2$$

$$= (1-\alpha)^t\mathbb{E}\|\boldsymbol{u}_0 - \boldsymbol{m}_0\|^2 + \alpha^2\sum_{s=1}^{t}(1-\alpha)^{t-s}\sigma^2.$$

Therefore we have

$$\mathbb{E}\|\boldsymbol{u}_t - \boldsymbol{m}_t\|^2 \leq \alpha\sigma^2.$$

So (15) becomes

$$\sum_{t=0}^{T-1}\left(\mathbb{E}\|\nabla f(\boldsymbol{x}_t) - \boldsymbol{u}_t\|^2 - (1-2L\gamma)\mathbb{E}\|\boldsymbol{u}_t\|^2\right)$$

$$\leq -\left(1 - \frac{2L^2\gamma^2}{\alpha^2} - \frac{4\gamma L}{\alpha} - 2L\gamma\right)\sum_{t=0}^{T-1}\mathbb{E}\|\boldsymbol{m}_t\|^2 + \left(\frac{2L^2\gamma^2}{\alpha^2} + \frac{3\gamma L}{2\alpha} + 4L\gamma\right)\sum_{t=0}^{T-1}\alpha\sigma^2$$

Therefore, for **Momentum SGD**

$$\sum_{t=0}^{T-1}A_t = \sum_{t=0}^{T-1}\mathbb{E}\|\nabla f(\boldsymbol{x}_t) - \boldsymbol{u}_t\|^2 - (1-2L\gamma)\mathbb{E}\|\boldsymbol{u}_t\|^2 - \frac{1}{4}\mathbb{E}\|\nabla f(\boldsymbol{x}_t)\|^2$$

$$\leq -\left(1 - \frac{2L^2\gamma^2}{\alpha^2} - \frac{4\gamma L}{\alpha} - 2L\gamma\right)\sum_{t=0}^{T-1}\mathbb{E}\|\boldsymbol{m}_t\|^2 + \left(\frac{2L\gamma}{\alpha} + \frac{3}{2} + 4\alpha\right)\gamma LT\sigma^2.$$

So if $\gamma L \leq \frac{\alpha}{12}$ and $\gamma L \leq \frac{1}{4}$, we have

$$\sum_{t=0}^{T-1} A_t \leq \frac{17\gamma L\sigma^2 T}{3}$$

$\square$

*Proof of Theorem 5.* For STROM, the most important part is to upper bound $\sum_{t=0}^{T-1} \mathbb{E}\|\boldsymbol{u}_t - \nabla f(\boldsymbol{x}_t)\|^2$. Therefore we first focus on this term.

Denoting $\boldsymbol{e}_t := \boldsymbol{u}_t - \nabla f(\boldsymbol{x}_t)$ and $\bar{\boldsymbol{g}}_t(\boldsymbol{x}) := \frac{1}{n}\sum_i \nabla F(\boldsymbol{x}; \xi_t^{(i)})$, we get

$\mathbb{E}\|\boldsymbol{e}_t\|^2$

$= \mathbb{E}\|(1-\alpha)\boldsymbol{e}_{t-1} + (\bar{\boldsymbol{g}}_t(\boldsymbol{x}_t) - \nabla f(\boldsymbol{x}_t)) - (1-\alpha)(\bar{\boldsymbol{g}}_t(\boldsymbol{x}_{t-1}) - \nabla f(\boldsymbol{x}_{t-1}))\|^2$

$= \mathbb{E}\|(1-\alpha)\boldsymbol{e}_{t-1}\|^2 + \mathbb{E}\|(\bar{\boldsymbol{g}}_t(\boldsymbol{x}_t) - \nabla f(\boldsymbol{x}_t)) - (1-\alpha)(\bar{\boldsymbol{g}}_t(\boldsymbol{x}_{t-1}) - \nabla f(\boldsymbol{x}_{t-1}))\|^2$

$= (1-\alpha)^2 \mathbb{E}\|\boldsymbol{e}_{t-1}\|^2 + \mathbb{E}\|\alpha(\bar{\boldsymbol{g}}_t(\boldsymbol{x}_t) - \nabla f(\boldsymbol{x}_t)) + (1-\alpha)(\bar{\boldsymbol{g}}_t(\boldsymbol{x}_t) - \bar{\boldsymbol{g}}_t(\boldsymbol{x}_{t-1}) + \nabla f(\boldsymbol{x}_{t-1}) - \nabla f(\boldsymbol{x}_t))\|^2$

$\leq (1-\alpha)^2 \mathbb{E}\|\boldsymbol{e}_{t-1}\|^2 + 2\alpha^2 \mathbb{E}\|\bar{\boldsymbol{g}}_t(\boldsymbol{x}_t) - \nabla f(\boldsymbol{x}_t)\|^2 + 2(1-\alpha)^2 \mathbb{E}\|\bar{\boldsymbol{g}}_t(\boldsymbol{x}_t) - \bar{\boldsymbol{g}}_t(\boldsymbol{x}_{t-1}) + \nabla f(\boldsymbol{x}_{t-1}) - \nabla f(\boldsymbol{x}_t)\|^2$

$= (1-\alpha)^2 \mathbb{E}\|\boldsymbol{e}_{t-1}\|^2 + 2\alpha^2 \mathbb{E}\|\bar{\boldsymbol{g}}_t(\boldsymbol{x}_t) - \nabla f(\boldsymbol{x}_t)\|^2$

$\qquad + \frac{2(1-\alpha)^2}{n^2}\mathbb{E}\left\|\sum_{i=1}^n \left(\nabla F(\boldsymbol{x}_t; \xi_t^{(i)}) - \nabla f_i(\boldsymbol{x}_t) - \nabla F(\boldsymbol{x}_{t-1}; \xi_t^{(i)}) + \nabla f_i(\boldsymbol{x}_{t-1})\right)\right\|^2$

$= (1-\alpha)^2 \mathbb{E}\|\boldsymbol{e}_{t-1}\|^2 + 2\alpha^2 \mathbb{E}\|\bar{\boldsymbol{g}}_t(\boldsymbol{x}_t) - \nabla f(\boldsymbol{x}_t)\|^2$

$\qquad + \frac{2(1-\alpha)^2}{n^2}\sum_{i=1}^n \mathbb{E}\left\|\nabla F(\boldsymbol{x}_t; \xi_t^{(i)}) - \nabla f_i(\boldsymbol{x}_t) - \nabla F(\boldsymbol{x}_{t-1}; \xi_t^{(i)}) + \nabla f_i(\boldsymbol{x}_{t-1})\right\|^2$

$\leq (1-\alpha)^2 \mathbb{E}\|\boldsymbol{e}_{t-1}\|^2 + 2\alpha^2 \mathbb{E}\|\bar{\boldsymbol{g}}_t(\boldsymbol{x}_t) - \nabla f(\boldsymbol{x}_t)\|^2$

$\qquad + \frac{4(1-\alpha)^2}{n^2}\sum_{i=1}^n \mathbb{E}\left\|\nabla F(\boldsymbol{x}_t; \xi_t^{(i)}) - \nabla F(\boldsymbol{x}_{t-1}; \xi_t^{(i)})\right\|^2 + \frac{4(1-\alpha)^2}{n^2}\sum_{i=1}^n \mathbb{E}\|\nabla f_i(\boldsymbol{x}_t) - \nabla f_i(\boldsymbol{x}_{t-1})\|^2$

$\leq (1-\alpha)^2 \mathbb{E}\|\boldsymbol{e}_{t-1}\|^2 + \frac{2\alpha^2\sigma^2}{n} + \frac{4(1-\alpha)^2(L^2 + L_F^2)}{n}\mathbb{E}\|\boldsymbol{x}_t - \boldsymbol{x}_{t-1}\|^2$

$\leq (1-\alpha)^2 \mathbb{E}\|\boldsymbol{e}_{t-1}\|^2 + \frac{2\alpha^2\sigma^2}{n} + \frac{4(1-\alpha)^2(L^2 + L_F^2)\gamma^2}{n}\mathbb{E}\|\boldsymbol{u}_{t-1}\|^2. \qquad (16)$

Using Lemma 9, we get

$$\sum_{t=0}^{T-1} \mathbb{E}\|\boldsymbol{e}_t\|^2 \leq \frac{\sum_{t=0}^{T-1}\left(\frac{2\alpha^2\sigma^2}{n} + \frac{4(1-\alpha)^2(L^2+L_F^2)\gamma^2}{n}\mathbb{E}\|\boldsymbol{u}_{t-1}\|^2\right) + \mathbb{E}\|\boldsymbol{u}_0 - \nabla f(\boldsymbol{x}_0)\|^2}{1-(1-\alpha)^2}$$

$$\leq \frac{1}{\alpha}\left(\frac{2\alpha^2\sigma^2 T}{n} + \frac{4(1-\alpha)^2(L^2+L_F^2)\gamma^2}{n}\sum_{t=0}^{T-1}\mathbb{E}\|\boldsymbol{u}_{t-1}\|^2\right) + \frac{\mathbb{E}\|\boldsymbol{u}_0 - \nabla f(\boldsymbol{x}_0)\|^2}{\alpha}$$

$$= \frac{2\alpha\sigma^2 T}{n} + \frac{4(1-\alpha)^2(L^2+L_F^2)\gamma^2}{\alpha n}\sum_{t=0}^{T-1}\mathbb{E}\|\boldsymbol{u}_{t-1}\|^2 + \frac{\mathbb{E}\|\boldsymbol{u}_0 - \nabla f(\boldsymbol{x}_0)\|^2}{\alpha}. \qquad (17)$$

Since $A_t = \mathbb{E}\|\nabla f(\hat{\boldsymbol{x}}_t) - \hat{\boldsymbol{u}}_t\|^2 - (1-2L\gamma)\mathbb{E}\|\hat{\boldsymbol{u}}_t\|^2 - \frac{\mathbb{E}\|\nabla f(\hat{\boldsymbol{x}}_t)\|^2}{4}$, therefore if $\frac{4(1-\alpha)^2(L^2+L_F^2)\gamma^2}{\alpha n} \leq \frac{1}{2}$ and $1-2L\gamma \geq \frac{1}{2}$, then we have

$$\sum_{t=0}^{T-1} A_t \leq \frac{2\alpha\sigma^2 T}{n} + \frac{\mathbb{E}\|\boldsymbol{u}_0 - \nabla f(\boldsymbol{x}_0)\|^2}{\alpha}$$

$$\leq \frac{2\alpha\sigma^2 T}{n} + \frac{\sigma^2}{n\alpha B_0}.$$

$\square$

*Proof of Theorem 7.* Here we introduce an auxiliary variable that is defined as
$$Z(\boldsymbol{x}, \boldsymbol{y}) := \nabla f(\boldsymbol{x}) - \nabla f(\boldsymbol{y}) - \langle \nabla^2 f(\boldsymbol{y}), \boldsymbol{x} - \boldsymbol{y} \rangle.$$
Then from the bounded Hessian in Assumption 3, we have
$$\|Z(\boldsymbol{x}, \boldsymbol{y})\| \leq \rho \|\boldsymbol{x} - \boldsymbol{y}\|^2.$$
Moreover, since $\boldsymbol{u}_t$ is a weighted average over all history stochastic gradients, under the bounded stochastic gradient of Assumption 3, we have
$$\mathbb{E}\|\boldsymbol{x}_t - \boldsymbol{x}_{t-1}\|^2 \leq \gamma^2 \Delta^2.$$
Denote $\boldsymbol{e}_t := \boldsymbol{v}_t - \nabla f(\boldsymbol{x}_t)$, we get

$\mathbb{E}\|\boldsymbol{e}_t\|^2$

$$= \mathbb{E}\left\| (1-\alpha)\boldsymbol{e}_{t-1} + \alpha \nabla F\left( \boldsymbol{x}_t + \frac{1-\alpha}{\alpha}(\boldsymbol{x}_t - \boldsymbol{x}_{t-1}); \xi_t^{(i)} \right) - \nabla f(\boldsymbol{x}_t) + (1-\alpha)\nabla f(\boldsymbol{x}_{t-1}) \right\|^2$$

$$= \mathbb{E}\left\| (1-\alpha)\boldsymbol{e}_{t-1} + \alpha \nabla f\left( \boldsymbol{x}_t + \frac{1-\alpha}{\alpha}(\boldsymbol{x}_t - \boldsymbol{x}_{t-1}) \right) - \nabla f(\boldsymbol{x}_t) + (1-\alpha)\nabla f(\boldsymbol{x}_{t-1}) \right\|^2$$

$$\quad + \mathbb{E}\left\| \alpha \nabla F\left( \boldsymbol{x}_t + \frac{1-\alpha}{\alpha}(\boldsymbol{x}_t - \boldsymbol{x}_{t-1}); \xi_t^{(i)} \right) - \alpha \nabla f\left( \boldsymbol{x}_t + \frac{1-\alpha}{\alpha}(\boldsymbol{x}_t - \boldsymbol{x}_{t-1}) \right) \right\|^2$$

$$\leq \mathbb{E}\left\| (1-\alpha)\boldsymbol{e}_{t-1} + \alpha \nabla f\left( \boldsymbol{x}_t + \frac{1-\alpha}{\alpha}(\boldsymbol{x}_t - \boldsymbol{x}_{t-1}) \right) - \nabla f(\boldsymbol{x}_t) + (1-\alpha)\nabla f(\boldsymbol{x}_{t-1}) \right\|^2 + \frac{\alpha^2 \sigma^2}{n}$$

$$= \mathbb{E}\left\| (1-\alpha)\boldsymbol{e}_{t-1} + (1-\alpha)Z(\boldsymbol{x}_{t-1}, \boldsymbol{x}_t) + \alpha Z\left( \boldsymbol{x}_t + \frac{1-\alpha}{\alpha}(\boldsymbol{x}_t - \boldsymbol{x}_{t-1}), \boldsymbol{x}_t \right) \right\|^2 + \frac{\alpha^2 \sigma^2}{n}$$

$$\leq (1+\alpha)\,\mathbb{E}\left\| (1-\alpha)\boldsymbol{e}_{t-1} \right\|^2 + \left(1 + \frac{1}{\alpha}\right) \mathbb{E}\left\| (1-\alpha)Z(\boldsymbol{x}_{t-1}, \boldsymbol{x}_t) + \alpha Z\left( \boldsymbol{x}_t + \frac{1-\alpha}{\alpha}(\boldsymbol{x}_t - \boldsymbol{x}_{t-1}), \boldsymbol{x}_t \right) \right\|^2$$

$$\quad + \frac{\alpha^2 \sigma^2}{n} \qquad \text{(applying } (x+y)^2 \leq (1+\alpha)x^2 + (1+\frac{1}{\alpha})y^2)$$

$$\leq (1-\alpha)\,\mathbb{E}\left\| \boldsymbol{e}_{t-1} \right\|^2 + \frac{(\alpha+1)}{\alpha^2}\mathbb{E}\left\| (1-\alpha)Z(\boldsymbol{x}_{t-1}, \boldsymbol{x}_t) \right\|^2$$

$$\quad + \frac{(\alpha+1)}{\alpha(1-\alpha)}\mathbb{E}\left\| \alpha Z\left( \boldsymbol{x}_t + \frac{1-\alpha}{\alpha}(\boldsymbol{x}_t - \boldsymbol{x}_{t-1}), \boldsymbol{x}_t \right) \right\|^2 + \frac{\alpha^2 \sigma^2}{n}$$

$$\leq (1-\alpha)\,\mathbb{E}\left\| \boldsymbol{e}_{t-1} \right\|^2 + \frac{(1-\alpha^2)(1-\alpha)\rho^2}{\alpha^3}\mathbb{E}\left\| \boldsymbol{x}_{t-1} - \boldsymbol{x}_t \right\|^4 + \frac{\alpha^2 \sigma^2}{n}$$

$$\leq (1-\alpha)\,\mathbb{E}\left\| \boldsymbol{e}_{t-1} \right\|^2 + \frac{(1-\alpha)\rho^2}{\alpha^3}\mathbb{E}\left\| \boldsymbol{x}_{t-1} - \boldsymbol{x}_t \right\|^4 + \frac{\alpha^2 \sigma^2}{n}. \tag{18}$$

In this case, we also have $\mathbb{E}_{\xi_{t-1}}\|\boldsymbol{x}_t - \boldsymbol{x}_{t-1}\|^2 \leq \gamma^2 \Delta^2$, and (18) becomes
$$\mathbb{E}\|\boldsymbol{e}_t\|^2 \leq (1-\alpha)\,\mathbb{E}\|\boldsymbol{e}_{t-1}\|^2 + \frac{(1-\alpha)\rho^2\gamma^4\Delta^4}{\alpha^3} + \frac{\alpha^2\sigma^2}{n}. \tag{19}$$
Using Lemma 9 and (19), we get
$$\sum_{t=0}^{T}\mathbb{E}\|\nabla f(\boldsymbol{x}_t) - \boldsymbol{v}_t\|^2 \leq \frac{\sum_{t=1}^{T}\left( \frac{\alpha^2\sigma^2}{n} + \frac{\rho^2\gamma^4\Delta^4}{\alpha^3} \right) + \mathbb{E}\|\boldsymbol{e}_0\|^2}{1 - (1-\alpha)} \leq \frac{\alpha\sigma^2 T}{n} + \frac{\sigma^2}{\alpha n B_0} + \frac{\rho^2\gamma^4\Delta^4 T}{\alpha^4}.$$
The lemma is proved. $\qquad\square$

## 10 Proof of Corollary

### 10.1 Proof of Corollary 4

*Proof.* Combining Theorem 1 and 3 together, we shall get
$$\frac{1}{T}\sum_{t=0}^{T}\mathbb{E}\|\nabla f(\boldsymbol{x}_t)\|^2 \leq \frac{16(f(\boldsymbol{x}_0) - f^*)}{\gamma T} + \frac{136\gamma L\sigma^2}{3n} + (64L_{\mathscr{A}}^2 + 2L^2)\gamma^2\alpha^2 T\epsilon^2.$$

Since $L_{\mathscr{A}} = L$, after setting $\gamma = \min\left\{\frac{\alpha}{12L}, \frac{\sqrt{n}}{\sqrt{T}\sigma}, \left(\frac{1}{\epsilon^2 T}\right)^{\frac{1}{3}}\right\}$, it can be easily verified that we have

$$\frac{1}{T}\sum_{t=0}^{T}\mathbb{E}\|\nabla f(\boldsymbol{x}_t)\|^2 \leq \mathcal{O}\left(\frac{\sigma}{\sqrt{nT}} + \frac{\alpha^2}{(\epsilon T)^{\frac{2}{3}}} + \frac{1}{\alpha T}\right),$$

where we treat $f(\boldsymbol{x}_0) - f^*$ and $L$ as constants. $\qquad\square$

## 10.2 Proof of Corollary 6

*Proof.* Combining Theorem 1 and 5 together, and setting $\alpha = \frac{8L^2\gamma^2}{n}$ we shall get

$$\frac{1}{T}\sum_{t=0}^{T}\mathbb{E}\|\nabla f(\boldsymbol{x}_t)\|^2 \leq \frac{16(f(\boldsymbol{x}_0) - f^*)}{\gamma T} + \frac{128L^2\gamma^2\sigma^2}{n^2} + \frac{\sigma^2}{L^2\gamma^2 B_0 T} + \frac{64(64L_{\mathscr{A}}^2 + 2L^2)L^4\gamma^6\epsilon^2}{n}.$$

Since $L_{\mathscr{A}} = 2L$, after setting $\gamma = \min\left\{\frac{1}{4L}, \left(\frac{n^2}{\sigma^2 T}\right)^{\frac{1}{3}}, \left(\frac{n}{\epsilon^2 T}\right)^{\frac{1}{7}}\right\}$ and $B_0 = \frac{\sigma^{\frac{8}{3}}T^{\frac{1}{3}}}{n^{\frac{2}{3}}}$, then it can be easily verified that we have

$$\frac{1}{T}\sum_{t=0}^{T}\mathbb{E}\|\nabla f(\boldsymbol{x}_t)\|^2 \leq \mathcal{O}\left(\left(\frac{\sigma}{nT}\right)^{\frac{2}{3}} + \left(\frac{\epsilon^2}{nT^6}\right)^{\frac{1}{7}} + \frac{1}{T}\right),$$

where we treat $f(\boldsymbol{x}_0) - f^*$, $L$ as constants. $\qquad\square$

## 10.3 Proof of Corollary 8

*Proof.* Combining Theorem 1 and 7 together, we shall get

$$\frac{1}{T}\sum_{t=0}^{T}\mathbb{E}\|\nabla f(\boldsymbol{x}_t)\|^2 \leq \frac{16(f(\boldsymbol{x}_0) - f^*)}{\gamma T} + \frac{8\alpha\sigma^2}{n} + \frac{8\sigma^2}{\alpha n B_0 T} + \frac{8\rho^2\gamma^4\Delta^4}{\alpha^4} + \frac{8(64L_{\mathscr{A}}^2 + 2L^2)L^2\gamma^6\epsilon^2}{n}.$$

Since $L_{\mathscr{A}} = 2L$, after setting $\gamma = \min\left\{\frac{1}{2L}, \left(\frac{n^4}{\sigma^8 T^5}\right)^{\frac{1}{9}}, \left(\frac{n}{\epsilon^2 T}\right)^{\frac{1}{7}}\right\}$, $\alpha = \left(\frac{n^5}{\sigma^8 T^4}\right)^{\frac{1}{9}}$ and $B_0 = 1$, then it can be easily verified that we have

$$\frac{1}{T}\sum_{t=0}^{T}\mathbb{E}\|\nabla f(\boldsymbol{x}_t)\|^2 \leq \mathcal{O}\left(\left(\frac{\sigma^8}{n^4 T^4}\right)^{\frac{1}{9}} + \left(\frac{\epsilon^2}{nT^6}\right)^{\frac{1}{7}} + \frac{1}{T}\right),$$

where we treat $f(\boldsymbol{x}_0) - f^*$, $L$ and $\rho$ as constants. $\qquad\square$