# OpenReview forum: "ErrorCompensatedX: error compensation for variance reduced algorithms"
_NeurIPS.cc/2021/Conference — NeurIPS 2021 Poster_

### Official Review · Reviewer_7b4p · 2021-06-30

**Rating:** 6
**Confidence:** 4

**Summary:**

This paper proposes an extension to the error compensation framework to reduce communication burden in distributed systems. The idea is to use previous two (instead of one) accumulated compression errors in the history together with low-pass filter update. The extension is applied for variance reduced methods and is analyzed for non-convex smooth objectives. The work provides general framework to analyze such methods, it shows that asymptotic convergence of such methods is the same as without compression and error compensation. Two numerical experiments are included with linear regression and neural networks to illustrate effectiveness of the proposed extension.

**Limitations And Societal Impact:**

It might be a good idea to write a small section on limitations.

**Main Review:**

1. It is mentioned in the abstract (lines 8,9) that your analysis standard Error Compansation (EC) does not fully compensate the compression error. Where this statement is justified in your theory ?

2. Then, you propose ErrorCompansatedX (ECX), which presumably compensate the compression error fully and achieves the same rate as without compression. But the same claim about the rate is true for EC methods (see [1,4,5]).

3. Are the statements on reducing half (line 19) or 3% (line 24) of the communication volume without degrading the speed of convergence accurate ? Are they theoretical results for general training or numerical results on some datasets ?

4. Are the references of Momentum SGD in line 28 and STORM in line 29 accurate ? For example, STORM method is not from (Cutkosky & Mehta, 2020).

5. In the introduction (lines 32-34), it is claimed that *the best* compression method for Variance Reduced (VR) methods is found and it is the proposed ECX. Where exactly do you prove that it is the best one ? If so, it is the best one in what sense ?

6. The first point of contribution (lines 36-38) claims that ECX admits *faster convergence rate* compared to previous methods. However, section 4 (lines 145-146) says it gives the same (asymptotic) rate as uncompressed methods; something that previous methods also achieved (see [1,4,5]).

7. Line 38 mentions that ECX fully compensates all error history. What does this mean ? If you are referring the fact that ECX uses previous two errors, then what about using previous three or more error terms. If this statement is empirical and is based on some experiments (e.g., Figure 1), it should made more precise.

8. The second point of contribution (lines 39-44) decomposes the rate into the terms affected by compression and terms that will remain even with full precision training. I think this is typical for most of the compressed algorithms. Minor: equation (2) in line 44 is defined much later after line 90.

9. Lines 73-78 are confusing. Why you highlight linear rate from (Mishchenko et al., 2019; Liu et al., 2020) and (Horva´th et al., 2019) in strongly convex setup, when you consider non-convex objectives only and all those three papers also include some theory for non-convex optimization (which is not mentioned there) ?

10. EC update rule described after line 99 is not exactly accurate because of the step-size $\gamma$ (it should be inside the compression operator; see e.g. equation (8) of (Stich et al., 2018b) ).

11. Related to the claim made in line 106, see [3,6,7].

12. Is the dependence of $\epsilon$ in the rate of Corollary 4 accurate ? It is in the denominator, meaning that less compression leads to worse upper bound. Also, the same leading term is achieved by Doublesqueeze method for non-convex losses.

13. In the section for experiments it is mentioned about four implementations, but plots contain three. Assumption two contains the same expression twice.

14. At least 10 references do not have journal/conference names (if not published then usually arXiv version is being referenced).

Out of curiosity, why the method is called ErrorCompansatedX ?

---------------------------


What can be done to improve the paper ?

I1. It is claimed that the paper provides general theoretical analysis framework for error compensated and variance reduced methods. One can use that framework to design a new method, which would outperform current VR methods both theoretically and numerically in some (at least one) settings.

I2. By describing the improved behavior of some method, it is important to identify and highlight the aspect/compenent of the method which actually makes the improvement. For instance, the proposed Algorithm 1 adds two components on top of EC framework: low-pass filter and second previous compression error. Which component contributes to faster convergence in Figures 1 and 2 ? Are both of them needed ? Do we actually need second previous error, since as it was mentioned in lines 101-102, low-pass filter can be helpful for stabilizing the training and further improving the training speed.


I3. Avoid over-claimed or conflicting statements (see points 3,5,6 above)

I4. Fix the references and inaccuracies when citing prior work (see points 9,10,11,14)

---------------------------


[1] Hanlin Tang, Xiangru Lian, Chen Yu, Tong Zhang, Ji Liu. Doublesqueeze: Parallel stochastic gradient descent with double-pass error-compensated compression. ICML 2019.

[2] Horva´th, S., Kovalev, D., Mishchenko, K., Stich, S., & Richtarik, P.. Stochastic distributed learning with gradient quantization and variance reduction. arXiv 1904.05115, 2019.

[3] Xun Qian, Hanze Dong, Peter Richtarik, Tong Zhang. Error Compensated Loopless SVRG for Distributed Optimization. OPT2020: 12th Annual Workshop on Optimization for Machine Learning, 2020.

[4] Sebastian U. Stich, Sai Praneeth Karimireddy. The Error-Feedback Framework: Better Rates for SGD with Delayed Gradients and Compressed Updates. Journal of Machine Learning ResearchJMLR 21(237), 1–36, 2020.

[5] Sai Praneeth Karimireddy, Quentin Rebjock, Sebastian U. Stich, Martin Jaggi. Error Feedback Fixes SignSGD and other Gradient Compression Schemes, ICML 2019.

[6] Xun Qian, Peter Richtarik and Tong Zhang. Error Compensated Distributed SGD Can Be Accelerated, arXiv 2010.00091, 2020.

[7] Xun Qian, Hanze Dong, Peter Richtarik, Tong Zhang. Error Compensated Proximal SGD and RDA. OPT2020: 12th Annual Workshop on Optimization for Machine Learning, 2020.

**Time Spent Reviewing:**

7+7

---

> ### Author Response · Authors · 2021-08-10
> **Response to Review #4**
>
> We appreciate your constructive feedback. We will add more discussion about our contribution, algorithm design, experimental verfiication accodring to your suggestions in our revised version.
>
> + 1: It is mentioned in the abstract (lines 8,9) that your analysis standard Error Compansation (EC) does not fully compensate the compression error. Where this statement is justified in your theory ?
>     + At the beginning of section 3.2, we explained that the compression error would accumulate for the standard error compensation method. In addition, we prove that ECX admits a faster convergence speed than EC, as shown in Theorem 2. Only ECX could attain the same asymptotic convergence rate with non-compressed variants STORM and IGT. We will clarify this in our revised version.
>
> + 2: Then, you propose ErrorCompansatedX (ECX), which presumably compensate the compression error fully and achieves the same rate as without compression. But the same claim about the rate is true for EC methods (see [1,4,5]).
>     + The same claim about the rate for EC is only true for the SGD case. This is explained in Sec 3.2 as well (Line 123 when $\alpha=1$). However, ECX is a more general method that could achieve the same asymptotic convergence rate for both SGD and VR SGD, and EC is just one special case of ECX when $\alpha=1$. Our convergence result could directly reduce to the previous result for EC when setting $\alpha=1$, which proves the tightness of our bound.
> + 3: Are the statements on reducing half (line 19) or 3% (line 24) of the communication volume without degrading the speed of convergence accurate ? Are they theoretical results for general training or numerical results on some datasets?
>     +  Regarding the statements,  we provide both theoretical analysis (Theorem 2, 3, 5 and 7) and numerical results on some datasets studies for the verification. We choose 1-bit compression in the experiments therefore the communication is 3% of the original.
>
> + 4: Are the references of Momentum SGD in line 28 and STORM in line 29 accurate ? For example, STORM method is not from (Cutkosky & Mehta, 2020).
>     + Thanks for pointing it out. We apologize for this and will update the reference in our revised version. Also, we will check other references.
>
> + 5: In the introduction (lines 32-34), it is claimed that the best compression method for Variance Reduced (VR) methods is found and it is the proposed ECX. Where exactly do you prove that it is the best one ? If so, it is the best one in what sense?
>     + We did not prove it, but ECX actually outperforms all previous communication compression methods for VR algorithms. We will clarify this in our revised version.
>
> + 6: The first point of contribution (lines 36-38) claims that ECX admits faster convergence rate compared to previous methods. However, section 4 (lines 145-146) says it gives the same (asymptotic) rate as uncompressed methods; something that previous methods also achieved (see [1,4,5]).
>     + For SGD and Momentum SGD, previous methods proved EC admits the same asymptotic rate as the original algorithm when $\alpha$ is not very small. However, for recent VR algorithms STORM and IGT, where $\alpha$ needs to be very small, only ECX can achieve the same convergence speed with the original while EC runs much slower.
> + 7: Line 38 mentions that ECX fully compensates all error history. What does this mean ? If you are referring the fact that ECX uses previous two errors, then what about using previous three or more error terms. If this statement is empirical and is based on some experiments (e.g., Figure 1), it should made more precise.
>     + We prove that ECX admits the same asymptotic convergence rate as the original algorithm, even with communication compression, which cannot be achieved using the previous method (EC). We describe this achievement as *fully compensates all error history* as mentioned in previous work [Tang et. al., 2019]. Using only the previous **ONE** error can not achieve it. This claim is justified both theoretically (Theorems 1, 2, 3, 5, and 7) and empirically (see Figure 2).
>
> + 8: The second point of contribution (lines 39-44) decomposes the rate into the terms affected by compression and terms that will remain even with full precision training. I think this is typical for most of the compressed algorithms.
>     + We politely argue that this result is not trivial because error compensation is a non-Markov process, so that the convergence analysis is much more complicated. These general results provide an illuminative understanding of the convergence behavior of error compensation methods and can be easily applied to future VR algorithms.
>
> + 9: Lines 73-78 are confusing. Why you highlight linear rate from (Mishchenko et al., 2019; Liu et al., 2020) and (Horva´th et al., 2019) in strongly convex setup, when you consider non-convex objectives only and all those three papers also include some theory for non-convex optimization (which is not mentioned there)?
>     + As shown in Sec. 3.2, error compensation can not remove all the compression errors because the last compression error still exists.  Line 73-78 describes another approach that compresses the difference between the current gradient and momentum, and the compression error can approach zero if the difference approaches zero. This benefit works for deterministic cases because there is no stochastic from the gradient. For the SGD case, there is positive variance in the gradient, and the difference can not approach zero. We mention strongly convex is because previous work only provides the convergence rate of IGT for the strongly-convex case. We will explain it in the revision.
>
> + 10: EC update rule described after line 99 is not exactly accurate because of the step-size  (it should be inside the compression operator; see e.g. equation (8) of (Stich et al., 2018b) ).
>     + Actually, we adapt the updating rule from another paper, Tang et al., 2019 where they also used the EC strategy. Whether step size is inside or outside shall not impose any difference for the convergence speed.
>
> + 11: Related to the claim made in line 106, see [3,6,7].
>     + For SGD and Momentum SGD, previous methods proved EC admits the same asymptotic rate as the original algorithm when $\alpha$ is not very small. However, for recent VR algorithms STORM and IGT, where $\alpha$ needs to be very small, only ECX can achieve the same convergence speed with the original while EC runs much slower.
>
> + 12: Is the dependence of ϵ in the rate of Corollary 4 accurate ? It is in the denominator, meaning that less compression leads to worse upper bound.
>     + Thanks for your correction, actually $\epsilon$ should be on the numerator, we will update this in our revised version.
>
> + 12: the same leading term is achieved by Doublesqueeze method for non-convex losses.
>     + Yes, for momentum SGD, when you $\alpha$ is not very small, ECX has similar performance with EC.
>
> + 13: In the section for experiments it is mentioned about four implementations, but plots contain three.
>     + This is because the naive way of compression (without any error compensation) does not converge.
> + 14: At least 10 references do not have journal/conference names (if not published then usually arXiv version is being referenced).
>     + We will update this in our revised version and we appreciate your effort for checking.
>
> + 15: Out of curiosity, why the method is called ErrorCompansatedX ?
>     + We used this name because it adapts the idea from EC, but it is a more general and faster algorithm.
>
> + 16: Clarification of the contribution and format of the citations:
>     + We appreciate your efforts in checking our draft. For the contribution, we will clarify that our method focus on the momentum-type variance reduced algorithms and the scenario where ECX should be used. We will double-check the reference format in our revised version.
>
>
>
>
> + 17: By describing the improved behavior of some method, it is important to identify and highlight the aspect/compenent of the method which actually makes the improvement. For instance, the proposed Algorithm 1 adds two components on top of EC framework: low-pass filter and second previous compression error. Which component contributes to faster convergence in Figures 1 and 2 ? Are both of them needed ? Do we actually need second previous error, since as it was mentioned in lines 101-102, low-pass filter can be helpful for stabilizing the training and further improving the training speed.
>     + We appreciate your suggestions. Our experiments showed that for both EC and ECX, the low-pass filter is beneficial for stabilizing the training when using STORM and IGT. But only ECX can achieve a similar convergence rate with the original algorithm. We will add more ablation studies to show which factor is essential for accelerating the training after compression.
>
> + 18: Avoid over-claimed or conflicting statements (see points 3,5,6 above)
>     + Yes, we will make the update according to your suggestions in our revised version.
>
> + 19: Fix the references and inaccuracies when citing prior work (see points 9,10,11,14)
>     + Yes, we will make the update according to your suggestions in our revised version.
>
> ## Reference
> Hanlin Tang, Xiangru Lian, Chen Yu, Tong Zhang, Ji Liu. "DoubleSqueeze: Parallel Stochastic Gradient Descent with Double-Pass Error-Compensated Compression." ICML (2019).

---

> > ### Comment · Reviewer_7b4p · 2021-08-23
> > **Theory for IGT**
> >
> > Thanks the author(s) for their response. One more question.
> >
> >
> > > Only ECX could attain the same asymptotic convergence rate with non-compressed variants STORM and IGT.
> >
> > Could you explain this claim for the case of IGT ? What is the non-compressed rate of IGT and which work do you refer to ? (Arnold et. al., 2019) has very weak theory under very relaxed assumptions. Also the rate of IGT in your paper, as you mention after Corollary 8, is worse than the rate obtained for SGD. Could you comment on this result more, how it shows the theoretical power of ECX ?

---

> > > ### Author Response · Authors · 2021-08-27
> > > **Response to Review #4**
> > >
> > > To the best of our knowledge, there is no existing convergence rate for IGT under the *non-convex* case. Our IGT convergence rate analaysis for the nonconvex case follows the idea of Arnold et. al. (2019), which proves the convergence rate for *strongly convex* case. We agree that its rate might not be tight (no matter in strongly convex case or non-convex case). However, we want to emphasize that the goal of this work is to show how to apply the *error-compensated compression* to existing base algorithms in order to reduce the communication cost without hurting the original convergence efficiency. How to improve the convergence rate for the base algorithm is NOT the main purpose of our work. Please let us know if you know there is a better convergence rate for IGT. We believe that a similar improvement can be also indicated in ECX-IGT. Thanks for pointing out this. we will mention it in our final version.

---

> > > > ### Comment · Reviewer_7b4p · 2021-08-27
> > > > **Increase in score and further concerns**
> > > >
> > > > Thanks for the clarification. In that case, I think you would agree that the IGT part of the claim
> > > >
> > > > > Only ECX could attain the same asymptotic convergence rate with non-compressed variants STORM and IGT.
> > > >
> > > > *is exaggerated*. Of course, I was not asking to derive/improve the rate of vanilla IGT.
> > > >
> > > > ----------------
> > > >
> > > > *After reading the other reviews and author(s)' responses, I did another pass over the paper and realized that the author(s) clarified several concerns of my initial review.* **As a result, I am willing to increase my score to** ***"6: Marginally above the acceptance threshold"***. *However, there are a few points that abstain me increasing the score even further and recommending an accept.*
> > > >
> > > > - I still do not get why you devoted section 4.2.3 on IGT. How it improves the presentation and theoretical importance of ECX ? For now, ECX on IGT recovers *some* rate, which is worse than for SGD. Besides, we do not know tight rate of IGT in non-convex case, thus we cannot really evaluate the addition of ECX on top it.
> > > > For SGD (with and without momentum), ECX gives the same rate, which is a nice property to have. However, ECX is not needed in this case. *To me, the only useful application of ECX, which gives an essential improvement is the application on STORM method.*
> > > >
> > > > - My second point is about the importance of the two components of ECX over standard EC. As far as I understood, ECX algorithmically combines previous compression error (standard EC), $2^{nd}$ previous compression error and low-pass filter. It is mentioned in the paper that adding low-pass filter is beneficial from practical side. But is it really necessary to get the theoretical results ? In other words, do you really need to include both components ($2^{nd}$ previous compression error and low-pass filter) into ECX to guarantee the improved rates ? I think this kind of ablation studies are necessary both in theory and in experiments.
> > > >
> > > > - Lastly, experimental section is quite limited. I agree with the *Reviewer HwbF* on this point to include numerical results for a few compressors/datasets/models and some experiments showing communication efficiency (time or bit comparisons).

---

### Official Review · Reviewer_LLeQ · 2021-07-14

**Rating:** 6
**Confidence:** 3

**Summary:**

This paper introduces a new method for error compensation in distributed optimization algorithms, specifically compressed gradient descent-based (CGD) algorithms, with the ultimate goal of reducing communication cost. Compared to existing error compensation approaches, the proposed method, termed ErrorCompensatedX, utilizes the error from the previous two steps of compressed gradient descent to improve it’s convergence rate. Despite some promising theoretical results, the numerical experiments seem inadequate, and below average exposition makes parts of the paper difficult to fully comprehend.

**Limitations And Societal Impact:**

Discussion regarding limitations of the proposed method seems lacking.

Paper is mainly theoretical and societal impact depends on how the practitioner would deploy the proposed method.

**Main Review:**

While the idea presented seems promising and some theoretical results are provided, this paper does not clearly indicate whether the proposed scheme reduces communication cost. Exposition is suboptimal and quite dense in places, and as such makes judging the paper on its’ merits and identifying novelties difficult.

Some further comments are below:
1. Since this paper concerns communication efficient optimization, one would expect to see theoretical and empirical results regarding communication complexity, communication rounds and number of communicated bits. The empirical results presented in Sec. 5 focus only on training loss and testing accuracy versus the number of training epochs.
2. Is $\xi^{(i)}$ in Eq. (1) different from $\xi_t$ in Eq. (2)?
3. Is $\alpha_t$ a positive scalar, or can it take any value?
4. Some additional intuition regarding the role of $\mathbf{e}_t$ in the equations after line 103, could facilitate better understanding of the proposed method. Also, how  does this equation describe a low-pass filter?
5. In the beginning of Section 3.2 it is mentioned that $\mathscr{A}$ is transferred from worker nodes to the server, however the ensuing equations do not involve the workers. The note in line 126 that the discussion focused on the one worker case should be mentioned earlier, ideally in the beginning of the subsection.
6. The difference between $\mathbf{\delta}_s$ between the second and third equations in Sec. 3.2 can be confusing.
7. The innovation of ErrorCompensatedX compared to regular Error Compensation should be made more explicit in Sec. 3.2.
8. In line 122 it is mentioned that when $\alpha = 1$, the last term disappears. However, if $\alpha = 1$ all terms except $\mathbf{x}_0$ seem to disappear.
9. In line 133 it is mentioned that "using just the compression error from the last step is not good enough". Not good enough with respect to what criterion? Also, where is this analysis shown?
10. In Assumption 2, the same inequality is written twice.
11. What does the learning rate in Sec. 5 refer to?


**Time Spent Reviewing:**

5

---

> ### Author Response · Authors · 2021-08-10
> **Response to Review #3**
>
> Thank you for your constructive feedback. We will add more details about the algorithm design and fix the typos in our updated version.
>
> + 1: Since this paper concerns communication efficient optimization, one would expect to see theoretical and empirical results regarding communication complexity, communication rounds and number of communicated bits. The empirical results presented in Sec. 5 focus only on training loss and testing accuracy versus the number of training epochs.
>     + We politely point out that our algorithm is a general framework for applying communication compression without degrading the convergence speed. We do not provide a convergence result w.r.t. the number of bits because different compression method admits different compression level. We prove that under the same communication budget, ErrorCompensatedX admits a faster convergence speed than previous communication compression methods, and our analysis strategy is widely used in previous work [Stich et al., 2018, Tang et al., 2019].
>
> + 2: Is $\xi^{(i)}$ in eq. (1) different from $\xi_t$ in eq. (2)?
>     + Yes. eq. (2) considers the single-thread version, and $\xi_t$ is one $\xi^{(i)}$ in eq. (1).
>
>
> + 3: Is $\alpha_t$ a positive scalar, or can it take any value?
>
>     + Yes $\alpha_t$ is a positive number and less than 1.
>
> + 4: Some additional intuition regarding the role of $e_t$ in the equations after line 103, could facilitate better understanding of the proposed method. Also, how does this equation describe a low-pass filter?
>
>     + The difference between this equation and the previous equation without low-pass filter is the introduction of $e_t$. When $\beta=1$, there is no low-pass filter and they are the same. When $\beta<1$, we add back the history compression error as well onto $g_t$. It helps reduce the variance of the compression error being added back, and the low-pass filter is used in [Chen et al., 2020; Wu et al., 2018] to describe it.
>
> + 5: In the beginning of Section 3.2 it is mentioned that $\mathcal{A}$ is transferred from worker nodes to the server, however the ensuing equations do not involve the workers. The note in line 126 that the discussion focused on the one worker case should be mentioned earlier, ideally in the beginning of the subsection.
>
>     + Thanks for the suggestion. We will make it clear in the revision by moving the assumption to the beginning of the subsection and adding more details.
>
> + 6: The difference between $\delta$s between the second and third equations in Sec. 3.2 can be confusing.
>     + In general we denote $\delta_s$ as the compression error. For the direct compression, the error is just $\delta_s$ from the compression. However, for the error compensation, the previous error $\delta_{s-1}$ that is not transferred is added back to the gradient before the compression. That's why there is a difference between $\delta_s$ and $\delta_{s-1}$.
>
> + 7: The innovation of ErrorCompensatedX compared to regular Error Compensation should be made more explicit in Sec. 3.2.
>     + In the first part of Sec. 3.2, we explained the motivation for our proposed ErrorCompensatedX. Regular error compensation will accumulate the error from compression (the last term in the equation between lines 121 and 122). Adding the additional term in ErrorCompensatedX removes the accumulated error from error compensation (the last term in the equation between lines 125 and 126). This is the major innovation of our proposed method.
>
> + 8: In line 122 it is mentioned that when $\alpha=1$, the last term disappears. However, if $\alpha=1$ all terms except $x_0$ seem to disappear.
>     + Note that we can have $s=t$ in the second term. Therefore when $\alpha=1$, the second term becomes $-\gamma\sum_{t=0}^{T-1}\mathcal{A}(x_t;\xi_t)$. Also, the last term becomes $\gamma \delta_s$ because $s$ can be $T-1$. Thus the compression error from the last step stays.
>
> + 9: In line 133 it is mentioned that "using just the compression error from the last step is not good enough". Not good enough with respect to what criterion? Also, where is this analysis shown?
>     + Here our result (both theoretical and empirical) showed that if using just one compression error (the standard error compression), we cannot ensure the same asymptotic convergence rate with the original algorithm, and the training speed would get slower. The theoretical analysis can be found in Theorem 2, and the experimental comparison is in Figure 2. We will clarify this in our revised version.
>
> + 10: In Assumption 2, the same inequality is written twice.
>     + You are right. Thanks for the comment. We will correct it in the revision.
>
> + 11: What does the learning rate in Sec. 5 refer to?
>     + This is $\gamma$ as described in Algorithm 1.
>
> ## Reference
>
> Stich, S. U., Cordonnier, J.-B., & Jaggi, M. "Sparsified sgd with memory." In Advances in Neural Information Processing Systems (2018).
>
> Hanlin Tang, Xiangru Lian, Chen Yu, Tong Zhang, Ji Liu. "DoubleSqueeze: Parallel Stochastic Gradient Descent with Double-Pass Error-Compensated Compression." ICML (2019).

---

> > ### Comment · Reviewer_LLeQ · 2021-09-01
> > **Response to author feedback**
> >
> > Thank you for your response and the clarifications.
> >
> > Based on these responses and the other reviewers comments, I am increasing the score to **"6: Marginally above the acceptance threshold"**, in the hopes that exposition and notation will be streamlined in the camera-ready version, to enhance the understanding of the proposed method.

---

### Official Review · Reviewer_56o9 · 2021-07-14

**Rating:** 7
**Confidence:** 3

**Summary:**

The paper proposes and analyse a generic error compensated scheme for variance reduced optimization algorithm in a distributed setting.

**Ethical Concerns:**

None.

**Limitations And Societal Impact:**

Yes.

**Main Review:**

The paper is well-written and pleasing to read. It is not excessively original, but the analysis is generic to several variance-reduced algorithms and very nicely carried out with a good balance between quantitative results and qualitative explanation. The contribution is also well-motivated with respect to previous work. The numerical experiments are also nicely presented without being excessively striking. It is the most appreciable paper in my batch of seven NeuRIPS papers this year.

**Time Spent Reviewing:**

4

---

> ### Author Response · Authors · 2021-08-10
> **Response to Review #2**
>
> Thanks for your positive comments. We appreciate your advice and will add more experiments in our revised version.

---

### Official Review · Reviewer_HwbF · 2021-07-15

**Rating:** 7
**Confidence:** 3

**Summary:**

In this paper, the authors present ErrorCompensatedX, which compensates the compression error using the information from the previous 2 steps instead of 1 for distributed SGD with variance reduction. Theoretical analysis is provided. Empirical results show that the proposed algorithm outperforms the baseline with single compensation.


**Limitations And Societal Impact:**

1. In some cases, such as momentum SGD, it seems that ErrorCompensatedX does not outperform single compensation.
2. Only 1 compressor (1-bit compressor) is tested in the experiments. Different compressors may have different behaviors. I also recommend to try top-k and random-k compressors.
3. Communication reduction is mostly for reducing the overall time for distributed training. However, no training time is provided in the experiments.


**Main Review:**

The paper is well-written. A novel method for improving the error compensation in variance-reduced SGD is proposed. Furthermore, a general theoretical analysis framework is provided for the proposed method. The experiment results also show good performance compared to the baseline. Overall, I think this is a good paper and I tend to accept it.
I have the following questions/concerns:
1. Compared to single compensation, how much extra computation and memory is cost by ErrorCompensatedX?
2. According to line 184-186, if alpha=1 (SGD case), single compensation and ErrorCompensatedX share the same error bound. Does that mean that in SGD case, single compensation is equivalent to ErrorCompensatedX? No matter if this is correct or not, could you give more explanation about such a special case?
3. I would recommend adding some more experiments. It could be either larger experiments like CIFAR-100/ImageNet, or applications other than CV, like NLP. Although I don’t think this is a big issue, using multiple datasets/applications in the experiments will be very helpful in showing the good performance of the proposed method.

**Time Spent Reviewing:**

4

---

> ### Author Response · Authors · 2021-08-10
> **Response to Review #1**
>
> Thank you for giving us constructive feedback. We will add more experiments and discussion according to your suggestions.
>
> + 1: Compared to single compensation, how much extra computation and memory is cost by ErrorCompensatedX?
>     + ErrorCompensatedX needs an extra buffer with the same size of the original gradient and two add operations for adding back the history compression error.
>
> + 2: According to line 184-186, if alpha=1 (SGD case), single compensation and ErrorCompensatedX share the same error bound. Does that mean that in SGD case, single compensation is equivalent to ErrorCompensatedX? No matter if this is correct or not, could you give more explanation about such a special case?
>     + Yes, when $\alpha=1$, the updating rule of ErrorCompensatedX is equivalent to single compensation, and they have the same error bound. It is because SGD does not use momentum to reduce the variance. We will clarify this in our revised version.
>
> + 3: I would recommend adding some more experiments. It could be either larger experiments like CIFAR-100/ImageNet, or applications other than CV, like NLP. Although I don’t think this is a big issue, using multiple datasets/applications in the experiments will be very helpful in showing the good performance of the proposed method.
>     + Thanks for your suggestion. We will add more experiments with large datasets and different applications in our revised version.
>
> + 4: In some cases, such as momentum SGD, it seems that ErrorCompensatedX does not outperform single compensation.
>     + Yes, as mentioned in our paper, for the case when  $\alpha$ is not very small, ErrorCompensatedX admits similar performance with Error compensation. We will make this limitation discussion more clear at the beginning of our paper.
>
> + 5: Additional experiments for more setting:
>     + Thanks for your suggestion. We will try different models and datasets in our revised version. We will also try different compression techniques such as top-k and random-k.
>
> + 6: Communication reduction is mostly for reducing the overall time for distributed training. However, no training time is provided in the experiments.
>     + The end-to-end training time speedup is highly dependent on the realization of the training hardware and the network condition. For your information, recent work [Tang et al., 2021] has already proved it using 1-bit compression. We will clarify this in our revised version.

---

> > ### Comment · Reviewer_HwbF · 2021-08-31
> > **response to authors' feedback**
> >
> > Thanks for the clarification. The authors answered my questions and resolved my concerns. I will keep my positive review.

---

### Decision · Program_Chairs · 2021-09-28

**Decision:**

Accept (Poster)

**Comment:**

I agree with the reviewers that this paper is well-written and has a nice analysis of several variance-reduced algorithms, balancing quantitative results and qualitative explanations. But I also agree with the reviewers that the exposition and notation could be improved, the effect of the low-pass filter could be better explained, and more extensive experiments could be carried out (especially for STORM, which has the best practical performance improvements, and for data sets where other values of alpha and beta are optimal). The reviewers give a large number of minor suggestions that can help you to significantly improve the paper.

I recommend that the paper should be accepted.

**Consistency Experiment:**

NeurIPS has a long history of experimentation. In 2014, NeurIPS ran an experiment in which 10% of submissions were reviewed by two independent committees to quantify the randomness in the review process. This year, we repeated a variant of this experiment to see how the quality of the review process has changed over time.  This paper was part of the experiment and was therefore assigned to two committees (consisting of reviewers, an Area Chair, and a Senior Area Chair) that reached independent decisions.  If both committees made the same recommendation, this recommendation was followed. If a single committee recommended acceptance, the paper was accepted (with the exception of a few cases in which the other committee identified what we considered a fatal flaw, e.g., an error in a key result).

This copy’s committee reached the following decision: **Accept (Poster)**

The other committee assigned to the paper recommended **Reject**.  You can find the other set of reviews, along with any follow up discussion with the authors here:
https://openreview.net/forum?id=fAWFaNaRVeF